# Immune surveillance of the lung by migrating tissue monocytes

**Mathieu P Rodero**[1,2,3†], **Lucie Poupel**[1,2,3†], **Pierre-Louis Loyher**[1,2,3†],
**Pauline Hamon**[1,2,3], **Fabrice Licata**[1,2,3], **Charlotte Pessel**[1,2,3], **David A Hume**[4],
**Christophe Combadière**[1,2,3], **Alexandre Boissonnas**[1,2,3*]

[1]Centre d'Immunologie et des Maladies Infectieuses, University Pierre et Marie Curie, Sorbonne Universities, Paris, France; [2]Centre d'Immunologie et des Maladies Infectieuses, Institut national de la santé et de la recherche médicale, U1135, Paris, France; [3]Centre d'Immunologie et des Maladies Infectieuses, Centre national de la recherche scientifique, ERL 8255, Paris, France; [4]Royal (Dick) School of Veterinary Studies, The Roslin Institute, Midlothian, United Kingdom

**Abstract** Monocytes are phagocytic effector cells in the blood and precursors of resident and inflammatory tissue macrophages. The aim of the current study was to analyse and compare their contribution to innate immune surveillance of the lung in the steady state with macrophage and dendritic cells (DC). ECFP and EGFP transgenic reporters based upon *Csf1r* and *Cx3cr1* distinguish monocytes from resident mononuclear phagocytes. We used these transgenes to study the migratory properties of monocytes and macrophages by functional imaging on explanted lungs. Migratory monocytes were found to be either patrolling within large vessels of the lung or locating at the interface between lung capillaries and alveoli. This spatial organisation gives to monocytes the property to capture fluorescent particles derived from both vascular and airway routes. We conclude that monocytes participate in steady-state surveillance of the lung, in a way that is complementary to resident macrophages and DC, without differentiating into macrophages.

*For correspondence: alexandre.boissonnas@upmc.fr

†These authors contributed equally to this work

Competing interests: The authors declare that no competing interests exist.

## Introduction

The mononuclear phagocytic system (MPS) consists of a family of cells—including monocytes, macrophages, and dendritic cells (DC)—that are derived from common committed bone marrow progenitors and perform related functions (*Hume, 2006*; *Hashimoto et al., 2011*). The lung is a mucosal surface of the body, exposed constantly to inhaled particles including pathogens, as well as other potential toxins (*Nelson et al., 2012*). Lung mononuclear phagocytes have been shown to adapt specifically to the lung environment, and contribute to lung homeostasis, scavenging, and immune surveillance (*Kopf et al., 2014*; *The lungs at the frontlines of immunity, 2014*). Monocytes were originally considered to be circulating precursors of macrophages, participating in renewal of tissue-resident macrophages in steady state and recruited in large numbers in response to inflammatory stimuli (*Auffray et al., 2009*; *Geissmann et al., 2010*). Fate mapping approaches and parabiosis experiments have been used to argue that tissue-resident macrophages such as alveolar macrophages (AM) may be seeded from the yolk sac or foetal liver during embryonic development and can be maintained in the absence of monocyte recruitment (*Guilliams et al., 2013*; *Hashimoto et al., 2013*; *Yona et al., 2013*). However, others have suggested the models used in these studies may disturb the availability of the key growth factor, CSF1, and do not necessarily reflect the steady state (*Jenkins and Hume, 2014*). Whatever their normal contribution to the resident macrophage pool, monocytes derived from the bone marrow (*Chow et al., 2011*) and spleen (*Swirski et al., 2009*) can

**eLife digest** White blood cells form part of the immune system, which protects the body against infectious diseases and other harmful agents. Some of these cells, including 'mononuclear phagocytes', can reside within different tissues of the body, such as the lungs. Other less specialized cells, called monocytes, circulate in the bloodstream. It had long been thought that once these monocytes had taken up residence in a tissue, they could only develop into tissue-resident phagocytes. Several researchers, however, recently reported that monocytes can also reside within tissues without becoming more specialized. Nevertheless, it remained unclear what these cells did when they were in these tissues.

Rodero, Poupel, Loyher et al. investigated the activities of tissue-resident monocytes found in the lungs of mice. First, mice were genetically engineered to produce fluorescent markers that meant that their monocytes could be easily distinguished from the mononuclear phagocytes in their lungs when viewed under a microscope. Rodero, Poupel, Loyher et al. then showed that the monocytes and the other mononuclear phagocytes localized to different regions of the lung. Further experiments showed that these two groups of cells also moved around the lungs in different ways. The tissue-resident monocytes surveyed both the blood vessels and airways, while the other tissue-resident mononuclear phagocytes only surveyed the airways.

These findings show that lung-resident monocytes perform a different role to those found in the bloodstream. The findings also open the way to improving our understanding of what tissue-resident monocytes do in other organs, and in healthy or diseased animals.

clearly replenish tissue macrophages after their cytotoxic depletion. Beyond this precursor role, monocytes carry out specific effector functions during infection (*Serbina et al., 2008*) and may be involved in steady-state tissue surveillance by capturing and transporting antigen from tissue to lymphoid organs (*Jakubzick et al., 2013*).

Macrophages isolated from different organs have distinct expression profiles which can be distinguished further if the cells are separated according to their surface markers (*Gautier et al., 2013*). Unfortunately, no available surface marker is well correlated with any other marker, at either the protein or mRNA level (*Hume, 2008, 2012*; *Hume et al., 2010*), so the number of macrophage subsets definable by flow cytometry is essentially infinite (*Becher et al., 2014*). Peripheral blood monocytes, on the other hand, can be subdivided into two broad functional classes. In the mouse, one subset called classical monocytes, expresses high levels of both Ly6C and the chemokine receptor CCR2 but a low level of the fractalkine (CX3CL1) receptor CX3CR1, while the second, the so-called non-classical monocytes, lacks Ly6C but expresses a high level of CX3CR1 (*Geissmann et al., 2003*).

In contrast to what we know about tissue macrophages, very little information is available on how monocytes behave after entering tissues. Myeloid-restricted fluorescent reporter genes based upon various lineage-restricted genes have been used in live imaging and functional genomics (*Hume, 2011*). A C*sf1r*-EGFP reporter gene serves as a definitive marker of MPS cells (*Sasmono et al., 2003*), while a *Cx3cr1*-EGFP reporter labels the non-classical monocytes as well as subsets of tissue macrophages, including microglia, as well as a subset of natural killer (NK) lymphocytes (*Jung et al., 2000*). The *Itgax*-YFP transgenic mouse was considered to provide a marker for classical 'dendritic cells' (*Lindquist et al., 2004*) and has been used to image so-called interstitial DC in the lung (*Looney et al., 2011*; *Thornton et al., 2012*). However, this reporter is rather uniformly expressed in tissue macrophages associated with mucosa and in the lung provides a generic MPS marker (*Hume, 2008, 2012*). The deletion of a conserved distal element of the *Csf1r* promoter in the *Csf1r-ECFP*$^{tg/+}$ mouse (MacBlue) ablates expression of a reporter gene in trophoblasts, osteoclasts, granulocytes, and many tissue macrophages (*Ovchinnikov et al., 2008*). This deleted promoter was used to construct an amplified binary transgene in which *Csf1r* promoter elements direct the expression of gal4-VP16, which in turn activates expression of a UAS-ECFP transgene. All blood monocytes in these MacBlue mice are strongly ECFP+, whereas most tissue macrophages do not express the reporter protein (*Sauter et al., 2014*). In the current study, we combined the myeloid-specific fluorescent reporters from MacBlue mice with either *Cx3cr1*$^{gfp/+}$ or *Itgax*-YFP transgenic mice to support in situ imaging of lung monocyte cell trafficking and compare their phagocytic activity with that of resident mononuclear phagocytes.

## Results

### The MacBlue×*Cx3cr1*$^{gfp/+}$ transgenic mouse discriminates lung mononuclear phagocyte subsets with specific tissue localization

The MacBlue binary transgene (*Csf1r*-gal4VP16/UAS-ECFP) provides a unique marker of blood monocytes (*Jenkins and Hume, 2014*). To confirm this restricted expression in the lung, we generated MacBlue×*Cx3cr1*$^{gfp/+}$ double transgenic mice. Two-photon laser scanning microscopy 3D-reconstruction of fresh explanted lung and histological section of cryo-preserved lungs from these mice identified distinct subsets with distinct morphologies and distributions within the organ (*Figure 1*). Stellar EGFP$^+$ECFP$^{neg}$ cells were seen in the collagen membrane surrounding the lung pleura (*Figure 1A,B*, green squares), deeper in the lung parenchyma with small round shapes or stellar shapes (*Figure 1B*, green squares), and along the basal membranes of bronchial airways (*Figure 1C*, green squares). The luminal side of airways and alveoli contained large round ECFP$^+$ cells, likely AM (*Figure 1A–D*, pink squares). Smaller amoeboid-like ECFP$^+$ cells were located in the interstitial space

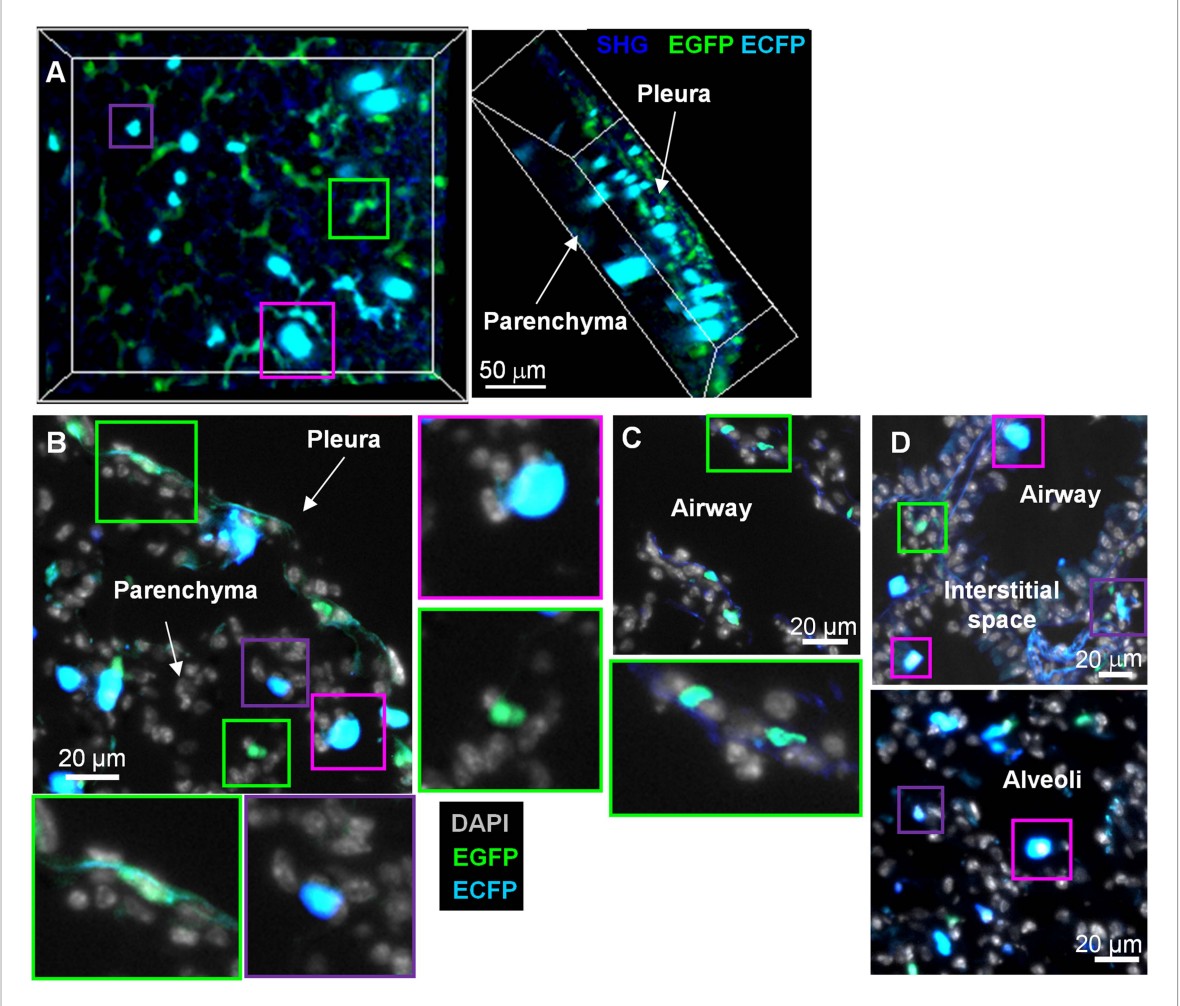

**Figure 1**. MacBlue×*Cx3cr1*$^{gfp/+}$ transgenic mouse discriminates lung mononuclear phagocyte subsets with specific tissue localization. **(A)** Front and side views of two-photon laser scanning microscopy (TPLSM) 3D reconstruction from pleura to alveolar space of explanted lung from a MacBlue×*Cx3cr1*$^{gfp/+}$ mouse. **(B)** MacBlue×*Cx3cr1*$^{gfp/+}$ mouse lung cryo-section showing lung pleura and parenchyma. **(C)** MacBlue×*Cx3cr1*$^{gfp/+}$ mouse lung cryo-section showing longitudinal view of bronchial airway. **(D)** MacBlue×*Cx3cr1*$^{gfp/+}$ mouse lung cryo-section showing interstitial space near bronchial airways and alveoli. Satellite images represent higher magnification of the corresponding coloured squares for each image. Images are representatives of more than three mice.

of the lung parenchyma (*Figure 1A–D*, purple squares). In overview, the pattern was consistent with previous evidence that the MacBlue ECFP transgene was expressed only in AM and monocyte-like cells, whereas most interstitial CX3CR1-EGFP expressing cells lacked expression.

## Differential fluorescent reporter expression discriminates mononuclear phagocyte subsets

To establish the relationship between the cells that could be imaged in situ and their cellular phenotypes in MacBlue×*Cx3cr1$^{gfp/+}$* mice, we applied a panel of phenotypic markers including CD11b, CD115, Ly6C, Ly6G, F4/80, CD64, CD11c, IAb, CD62L, NK1.1, and SiglecF to discriminate four different subsets of the lung based on their EGFP/ECFP signature in the double transgenic line (*Figure 2A*) and compared them to blood populations (*Figure 2B* and *Table 1*). The lungs contained two ECFP$^{bright}$ populations, either EGFP$^{bright}$ or EGFP$^{dim}$ (*Figure 2A*, purple gate) resembling those observed in the blood (*Figure 2B*). The CX3CR1-EGFP$^{low}$ population was Ly6C$^{high}$CD11b$^+$Ly6G$^{neg}$F4/80$^{int}$NK1.1$^{neg}$CD64$^+$ (blue gate) in both the blood and the lungs (*Table 1*), consistent with identity as 'classical' Ly6C$^{high}$ monocytes (Ly6C$^{high}$ Mo). The CX3CR1-EGFP$^{high}$ population was Ly6C$^{low}$CD11b$^+$Ly6G$^{neg}$F4/80$^{int}$NK1.1$^{neg}$CD64$^+$CD11c$^+$ phenotype (red gate), consistent with the phenotype of the Ly6C$^{low}$ monocyte subset (Ly6C$^{low}$ Mo) (*Guilliams et al., 2014*). For both subsets, the expression of CD115 and CD62L was down modulated in the lung cells compared to their circulating counterparts. Downregulation of surface CSF1R (CD115) could reflect the down-modulation of the surface receptor both by its ligand and by the many inflammatory stimuli present in the lung (*Sester et al., 1999*). As expected, the lung also contained an ECFP$^{high}$EGFP$^{neg}$ signature (pink gate). These cells were larger than monocytes and CD11b$^+$Ly6C$^{neg}$Ly6G$^{neg}$F4/80$^{high}$NK1.1$^{neg}$CD11c$^{high}$CD64$^{high}$SiglecF$^{high}$ cells, consistent with their identity as AM.

The EGFP$^{bright}$ cells that lacked detectable ECFP (green gate) were a heterogeneous population. A subset of these cells in the lung labelled with NK1.1, but many were CD64$^+$F4/80$^+$ (*Figure 2A* and *Table 1*) interstitial and pleural macrophages, as observed on histological reconstruction (*Figure 1A*). In the blood, the majority of ECFP$^{neg}$EGFP$^{bright}$ cells labelled with NK1.1 (*Figure 2B*), but the population also included immature myeloid cells as previously reported (*Sauter et al., 2014*). Note that the ECFP transgene is expressed at low but detectable levels on granulocytes in the blood (cyan gate); these cells were positive for Ly6G (*Figure 2B,C*). In summary, the binary expression of the two transgenic reporters in MacBlue×*Cx3cr1$^{gfp/+}$* mice permits the identification of monocytes in tissues, and their distinction from other mononuclear phagocyte subsets as well as from NK cells.

## Interstitial ECFP$^+$ cells are monocyte derived

In order to confirm the monocyte origin of the ECFP$^+$EGFP$^{low/high}$ -Ly6C$^{high}$ and -Ly6C$^{low}$ monocytes, we generated parabionts of the double transgenic MacBlue×*Cx3cr1$^{gfp/+}$* with C57Bl6 mice and analysed the reconstitution after 1 month of parabiosis. In the lung, donor-derived ECFP cells displayed Ly6C$^{high}$ and Ly6C$^{low}$ monocyte phenotypes (*Figure 3A*), were small in size, with ameboid-like morphologies, and were all located in the interalveolar space (*Figure 3B*). By contrast, neither AM nor interstitial macrophages were derived from the donor as there was no expression of the transgenes in these compartments (*Figure 3B*). The only cells expressing EGFP but not ECFP, were NK cells derived from the donor (*Figure 3A*).

## CCR2 and CX3CR1 control the accumulation of lung mononuclear phagocytes

To evaluate the function of chemotactic signals in the lung, we compared the frequency of the different subsets defined above in Ccr2$^{-/-}$, Cx3cr1$^{-/-}$, and Ccr2$^{-/-}$Cx3cr1$^{-/-}$ (dKO) mice in the blood and the lung (*Figure 4*). Consistent with published studies (*Kim et al., 2011*; *Yona et al., 2013*), the frequency of circulating Ly6C$^{high}$ monocytes was similar in wild-type (WT) and Cx3cr1$^{-/-}$ mice. By contrast, the absence of CX3CR1 did impact on the yield of Ly6C$^{high}$ cells in the lung. As expected, this subset was strongly reduced in Ccr2$^{-/-}$ mice and dKO mice in both the blood and the lungs (*Figure 4B,C*). Ly6C$^{low}$ monocyte frequency was similar in Ccr2$^{-/-}$ mice but significantly reduced in Cx3cr1$^{-/-}$ and dKO mice in both the blood and the lungs compared to WT mice (*Figure 4B,C*). AM frequencies were similar in all strains (*Figure 4C*) consistent with their reported monocyte-independent homeostasis (*Guilliams et al., 2013*). Together, these results showed that CCR2 and

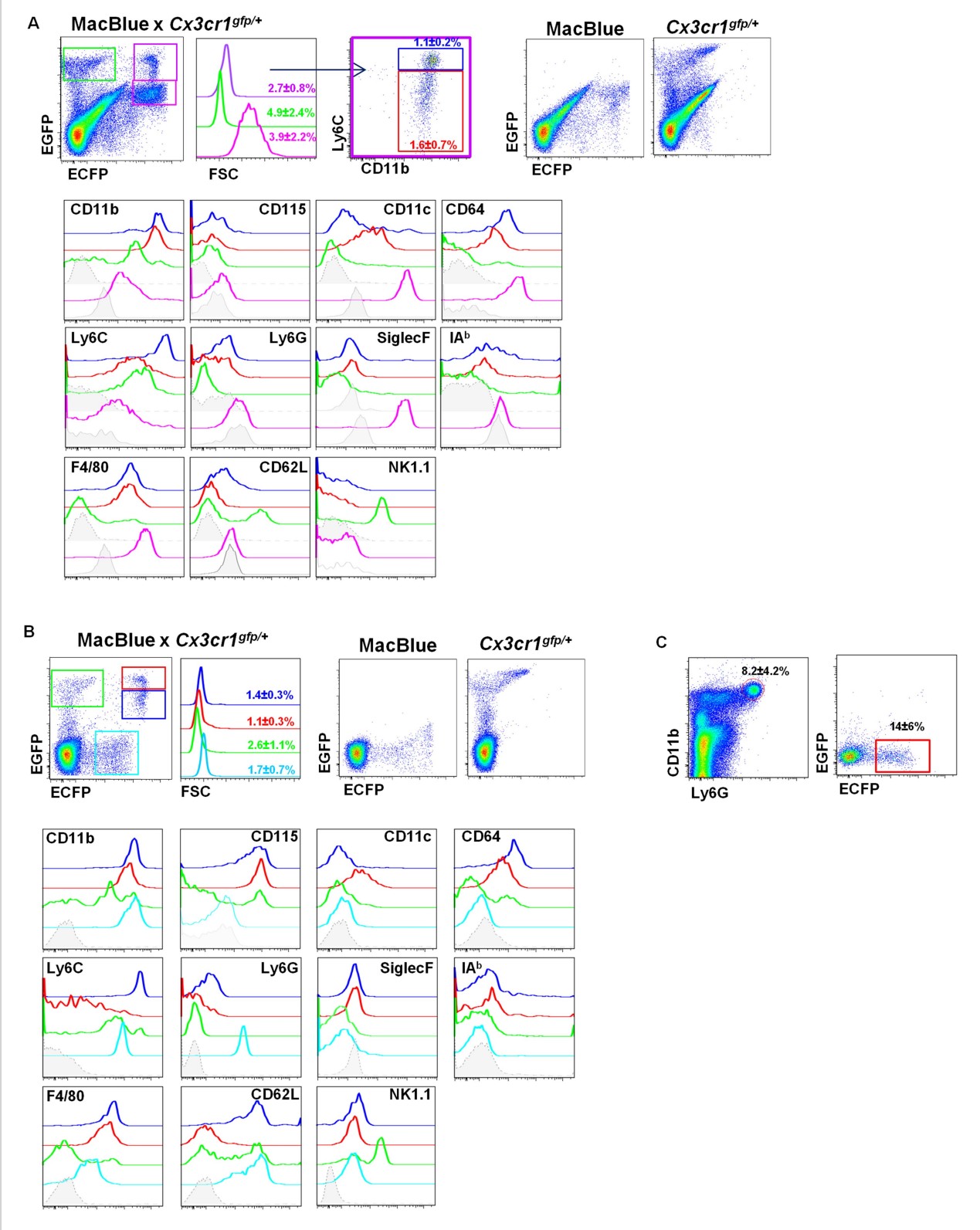

**Figure 2**. Differential fluorescent reporter expression discriminates mononuclear phagocyte subsets. ECFP and EGFP expression in **(A)** the lungs and **(B)** the blood of MacBlue×Cx3cr1$^{gfp/+}$ mice. Colour-coded gating identifies the main subsets according to EGFP/ECFP signature. Percentages ± SD of total cells according to colour code are indicated (n = 6 from two independent experiments). Dot plots showing spectral overlap of EGFP and ECFP fluorescence are depicted using separated MacBlue and Cx3cr1$^{gfp/+}$ transgenic mice. Overlay of histogram plots of indicated markers shows the

*Figure 2. continued on next page*

*Figure 2. Continued*

phenotype of the respective colour-coded gated cell populations. Grey histograms present the FMO (full minus one) signal gated on total monocytes. For lungs, lower grey histograms present the FMO signal gated on alveolar macrophages (AM) (pink gate). For the blood, the cyan subset expressing a low level of ECFP represents blood neutrophils. (C) Dot plots show the intensity and the frequency of ECFP expression on Ly6G$^+$ gated cells in the lungs. Means of percentage ± SD are indicated (n = 6 from two independent experiments).

CX3CR1 control the steady-state trafficking and survival of both Ly6C$^{high}$ and Ly6C$^{low}$ monocyte-derived cells in the lungs.

## Lung mononuclear phagocytes constitutively survey the entire space of the alveolar areas through distinct migratory patterns

Having demonstrated the utility of the MacBlue marker to define monocyte-derived cells, we performed live imaging on explanted lungs to characterize the behaviour of these cells (*Figure 5A* and *Video 1*). Small interstitial ECFP$^+$ cells (area 61 ± 17 μm) with an amoeboid-like shape displayed a bimodal behaviour: they were either highly motile, going backwards and forwards between the alveoli, or relatively sessile with protrusions extending towards the lumen of the alveoli (*Figure 5B–C* and *Videos 1 and 2*). Based upon these distinct behaviours, we defined 'patrolling' cells with high velocity (10 ± 4.6 μm/min) and a low arrest coefficient (14 ± 13%) and interstitial ECFP$^+$ cells with lower velocity (4 ± 2.2 μm/min) and a higher arrest coefficient (64 ± 21%) (*Figure 5C*). Intravenous injection of rhodamine dextran prior to mouse sacrifice permitted, during a short time frame before vascular leakage, to determine that fast moving patrolling ECFP$^+$ cells were located inside large vessels, whereas slow motile ECFP$^+$ cells appeared to be extravascular but in close contact with the vasculature (*Video 3*). Aside from monocytes, large round ECFP$^{bright}$ AM (area 121 ± 23 μm) were detected in the alveolar lumina, leisurely surveying the surface of the airways with a velocity of 2 ± 1.5 μm/min and a high arrest coefficient (76 ± 19%) (*Video 4*). Interstitial EGFP$^+$ cells also migrated slowly, with comparable velocity and arrest coefficient to AM (*Figure 5C*). To determine their motility coefficient (MC), we plotted the mean square displacement as a function of the square root of time. The MC of patrolling monocytes was 30-fold higher than the MC of interstitial ECFP$^+$ cells (39 vs 1.3 μm/min) and a further fourfold higher than the MC of AM (0.3 μm/min) (*Figure 5D*). This higher MC in interstitial ECFP$^+$ monocytes was attributable to the protrusive activity and slow displacement

**Table 1**. Comparative phenotype of mononuclear phagocyte (MP) subsets in the blood and the lung

| Tissue | Lung | Blood | Lung | Blood | Lung | Blood | Lung |
|---|---|---|---|---|---|---|---|
| CD11b | ++ | ++ | ++ | ++ | ++/+ | ++/+ | + |
| CD115 | − | ++ | − | ++ | −/− | +/− | − |
| Ly6C | ++ | ++ | +/− | +/− | +/+ | +/+ | − |
| Ly6G | − | − | − | − | − | − | − |
| F4/80 | + | + | + | + | ++/− | +/− | ++ |
| CD64 | + | + | + | + | ++/− | +/− | ++ |
| CD11c | − | − | + | + | +/− | +/− | ++ |
| IAb | +/− | +/− | +/− | +/− | +/− | −/− | − |
| CD62L | + | ++ | − | − | −/+ | +/+ | − |
| NK1.1 | − | − | − | − | −/++ | −/++ | − |
| SiglecF | − | − | − | − | −/− | −/− | ++ |
| Conclusion | Ly6C$^{high}$ Mo | | Ly6C$^{low}$ Mo | | Cx3cr1$^+$ MP/NK | | AM |

The relative expression of the markers for each subset in the blood and the lungs are compared and specific subsets defined according to the phenotype. ++ : high expression; + : positive expression; − : below FMO (full minus one) signal. AM, alveolar macrophages; NK, natural killer.

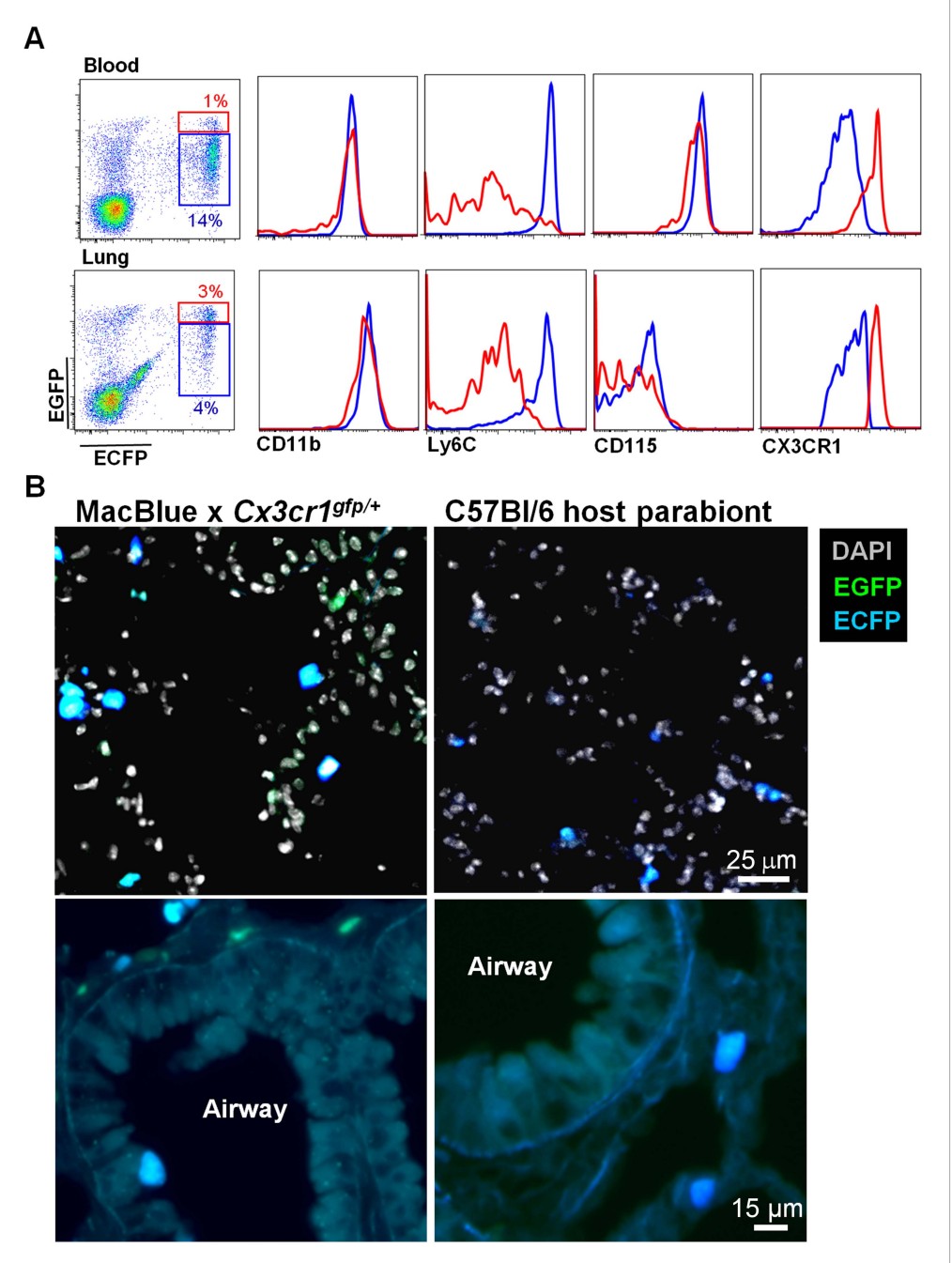

**Figure 3**. Interstitial ECFP[+] cells are monocyte-derived. **(A)** Dot plot shows the ECFP/EGFP chimerism in the blood and the lungs of C57Bl6 host parabiont with MacBlue×$Cx3cr1^{gfp/+}$ mouse. Histograms represent the expression of monocyte markers on the CX3CR1[low] (blue gate) and CX3CR1[high] cells (red gate). **(B)** Pictures compare different magnifications of lung cryo-section from MacBlue×$Cx3cr1^{gfp/+}$ mouse (left) with C57Bl6 host parabiont with MacBlue×$Cx3cr1^{gfp/+}$ mouse (right). Up to six parabionts were prepared independently.

suggesting scanning of the interalveolar space (*Video 2*). Time lapse imaging of explanted non-transgenic lungs from parabiont mice showed similar scanning behaviour (*Videos 5 and 6*), supporting the monocyte origin of these cells. In overview, while AM survey the luminal side of the alveoli, interstitial monocyte-derived cells survey the lung tissue through either active patrolling in the vasculature or protrusive activity toward the alveolar space.

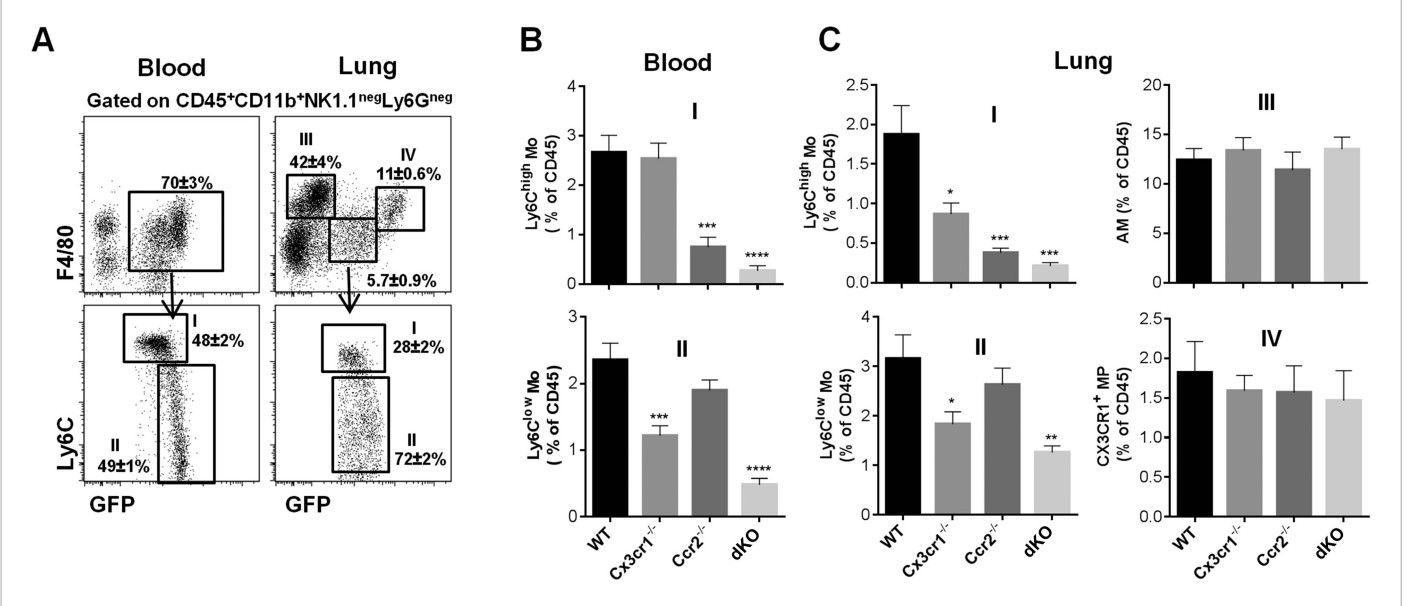

**Figure 4**. CCR2 and CX3CR1 control the accumulation of lung mononuclear phagocytes. **(A)** Gating strategy defines (I) Ly6C[high] monocytes, (II) Ly6C[low] monocytes, (III) alveolar macrophages (AM), and (IV) CX3CR1+ lung macrophages gated on CD45+CD11b+NK1.1[neg]Ly6G[neg] cells. Bars represent quantification as a percentage of CD45+ cells of the defined cell subsets from **(B)** the blood and **(C)** the lungs in *Cx3cr1*[gfp/+]*Ccr2*[rfp/+] (WT), *Cx3cr1*[gfp/gfp]*Ccr2*[rfp/+] (Cx3cr1[−/−]), *Cx3cr1*[gfp/+]*Ccr2*[rfp/-] (Ccr2[−/−]), and *Cx3cr1*[gfp/gfp]*Ccr2*[rfp/rfp] (dKO) mice. Bars represent means ± SEM (n = 10–13 mice per group from four independent experiments). ANOVA with Bonferroni adjustment was used. Mo, monocytes; MP, mononuclear phagocytes; WT, wild-type.

## Interalveolar space scanning by monocyte-derived cells is CX3CR1 dependent

In order to determine the molecular mechanism involved in the steady-state lung surveillance, we analysed the behaviour of lung mononuclear phagocytes in MacBlue×*Cx3cr1*[gfp/gfp] mice (*Figure 6A* and *Video 7*). The absence of functional CX3CR1 did not change the arrest coefficient (*Figure 6B*) but did slightly reduce the MC of patrolling cells (*Figure 6C*). As might be anticipated given their lack of CX3CR1 expression, AM behaviour was also comparable in WT and Cx3cr1[−/−] mice (*Figure 6B–C*). On the other hand, deletion of the CX3CR1 clearly altered the behaviour of the interstitial ECFP+ cells, which showed a greatly increased arrest coefficient and reduced MC (*Figure 6B–C*). The scanning behaviour was also strongly affected. In the absence of CX3CR1, the remaining interstitial ECFP+ cells present had greatly reduced protrusive activity, as indicated by increased global 'sphericity' (*Figure 6D,E*) and reduced sphericity variation (*Figure 6F*).

## Interstitial monocyte-derived cells localized in the alveolar space whereas lung dendritic cells preferentially localized near large airways

Lung DC defined by CD11c expression have been previously shown to participate in antigen uptake (*Thornton et al., 2012*). CD11c is expressed on all tissue macrophages in the lung, but is absent from monocytes. To compare the roles of these cells in particle clearance, we combined MacBlue with the *Itgax*-YFP transgene (MacBlue×*Itgax*-YFP) and again compared the fluorescent signatures of different mononuclear phagocyte populations (*Figure 7A*). According to previous phenotyping, Ly6C[high] monocytes were CD11b+Ly6G−SiglecF−IAb[low]CD64[low]Ly6C[high], Ly6C[low] monocytes were CD11b+Ly6G−SiglecF−IAb[low]CD64[low]Ly6C[low], interstitial macrophages were CD11b+Ly6G−SiglecF−IAb[high]CD64+, and AM were CD11b[low]CD11c[high]SiglecF[high]CD64+. Based upon published markers (*Guilliams et al., 2014*), two sets of classical myeloid DC have been defined: CD11b+ DC that were CD11b+CD11c+Ly6G−SiglecF−IAb[high]CD64−CD24+ and CD11b− DC that were CD11b−CD11c+Ly6G−SiglecF−IAb[high]CD64−CD24+CD103+ (*Figure 7A*). We determined the mean fluorescence intensity (MFI) of EGFP, ECFP, and YFP reporters from the MacBlue×*Cx3cr1*[gfp/+] and MacBlue×*Itgax*-YFP in these subsets

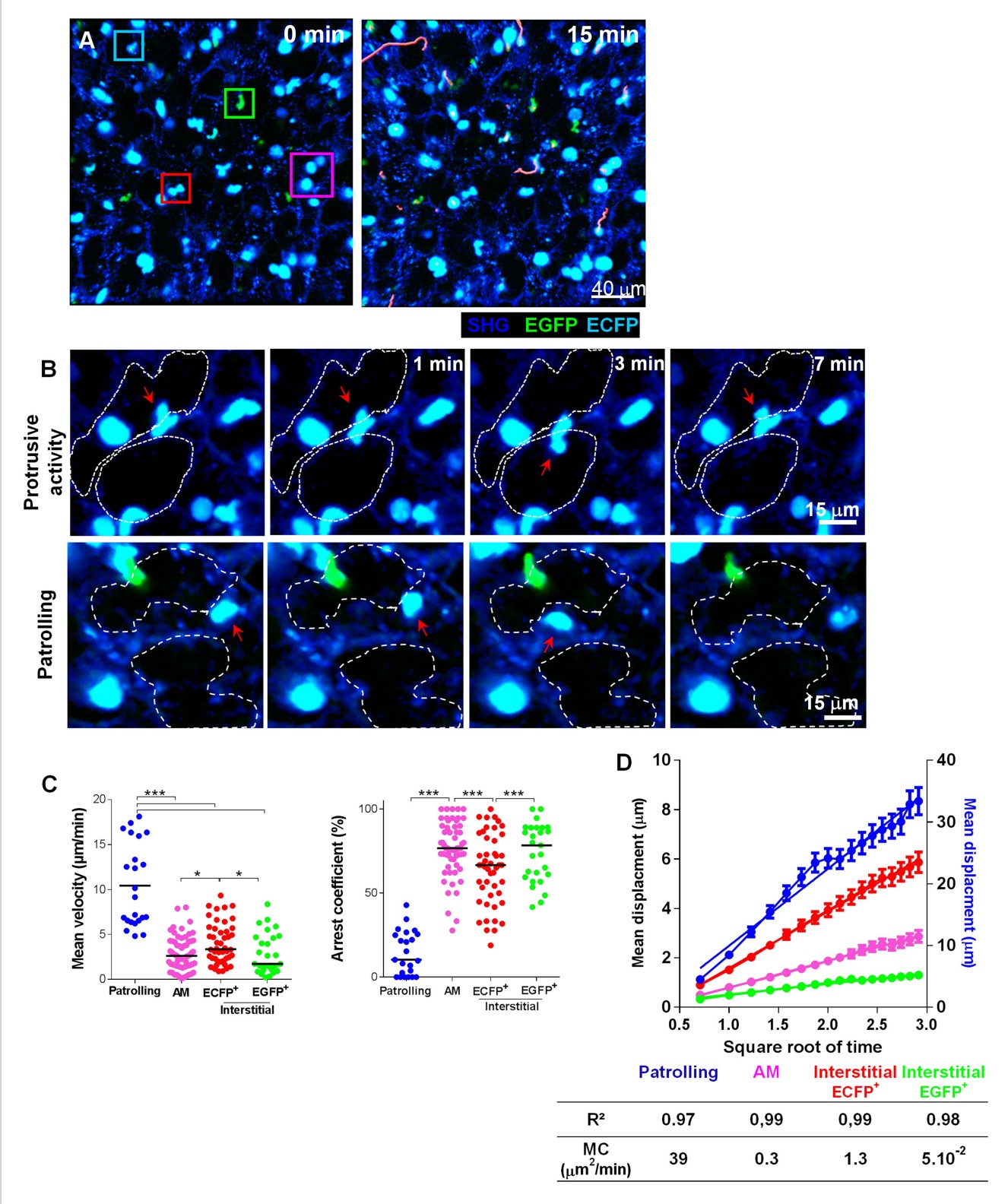

**Figure 5**. Lung mononuclear phagocytes constitutively survey the entire space of the alveolar areas through distinct migratory patterns. (A) Time series two-photon laser scanning microscopy (TPLSM) pictures with overlaid tracks of cell motility of the alveolar area from explanted lung. Pink, blue, red, and green squares surround alveolar macrophages, patrolling monocytes, interstitial ECFP+ cells, and interstitial EGFP+ cells, respectively. (B) Time series TPLSM pictures show representative protrusive activity (upper panel) and patrolling (lower panel) behaviour by interstitial ECFP+ cells. (C) Quantification of

*Figure 5. continued on next page*

*Figure 5. Continued*

the mean velocity and arrest coefficient of cell subsets. Bars indicate the medians. Data are pooled from six independent mice. Kruskal–Wallis tests followed by Dunn's multiple comparison tests were performed. **(D)** Mean displacement ± SEM as a function of the square root of time for alveolar macrophages (AM) (pink), scanning interstitial ECFP+ cells (red), EGFP+ cells (green) (left scale), and patrolling interstitial ECFP+ cells (blue) (right scale). Coloured lines represent the linear regression of the curves. $r^2$ and motility coefficients (MC = x/4t) are indicated.

(*Figure 7B*). EGFP and ECFP fluorescent reporter expressions were mainly restricted to lung macrophage and monocyte subsets as previously described (*Figure 2*). Although CD11c mRNA and surface protein are expressed widely among tissue macrophages in the lung, YFP expression was mainly restricted to the defined classical DC subsets with a brighter expression for CD11b− DC (*Figure 7C*). Interestingly, AM only express the MacBlue transgene and despite expression of CD11c protein on their surface, YFP was detectable in only a small subset (*Figure 7B–C*). Thus the combination of these different transgenes permits the distinction of lung mononuclear phagocytes and lung DC. AM represented the most abundant mononuclear phagocyte subset of the lung (*Figure 7D*). Lung monocytes represented the second main mononuclear phagocyte subset and were even more numerous than interstitial macrophages, CD11b+ DC and CD11b−CD103+ DC (*Figure 7D*). Histological analysis of a lung section of MacBlue×*Itgax*-YFP showed that YFP[bright] cells were mainly located along bronchial airways and poorly distributed in the alveolar space in contrast to ECFP+ monocytes and AM (*Figure 7E*).

## Lung monocyte-derived cells survey both airways and vascular routes, whereas DC survey only airways

Particle size and the physical properties of inhaled particles are critical in the uptake by lung phagocytes and trafficking to regional lymph nodes (*Jakubzick et al., 2008*; *Blank et al., 2013*). We hypothesized that regional segregation of monocytes and classical DC could be also important in antigen uptake. We inoculated mice with fluorescent beads either by intravenous or airway routes and compared the proportion of phagocytic cells in different lung subsets 4 hr after bead inoculation (*Figure 8A*). After i.v. inoculation, only monocytic cells appeared to have taken up particles (*Figure 8B–C*). After airway inhalation, AM were the main phagocytic subset (*Figure 8B–C*), but particles were also detected among interstitial cells. The numbers of phagocytic Ly6C[low] and Ly6C[high] monocytes were significantly higher compared to the number of phagocytic DC after airway inhalation, showing that lung monocytes are the only subsets that effectively capture both blood-derived and airway-derived fluorescent particles (*Figure 8C*).

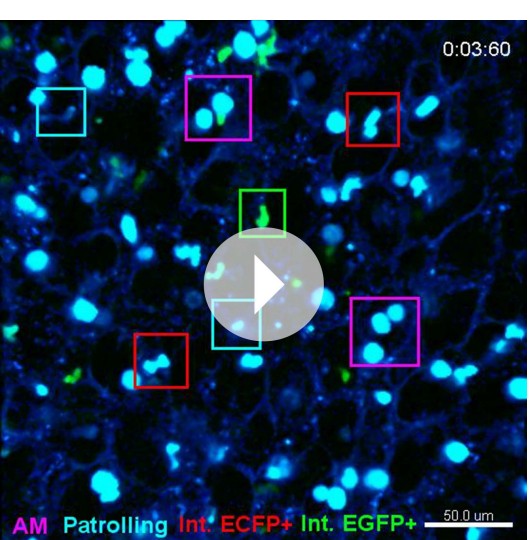

**Video 1.** Mononuclear phagocyte immune surveillance of alveolar space. Live 3D imaging of mononuclear phagocyte behaviour in the alveolar space of the lung of a MacBlue×*Cx3cr1*[gfp/+] mouse. The ECFP signal is in cyan, the EGFP signal in green, and the SHG signal (blue) indicates interstitial tissue and defines alveoli. Representative behaviour of alveolar macrophages (AM) (pink squares), patrolling ECFP+ cells (blue squares), interstitial ECFP+ cells (red squares), and interstitial EGFP+ cells (green square) are indicated.

## Lung monocyte-derived cells are located at the interface between blood and airways

In order to further discriminate lung monocytes from circulating monocytes, we performed blood/tissue partitioning using in vivo CD45 labelling as reported by others (*Anderson et al., 2014*; *Girgis et al., 2014*). This approach was reported to be technically challenging due to the vascular permeability of the lung (*Jakubzick*

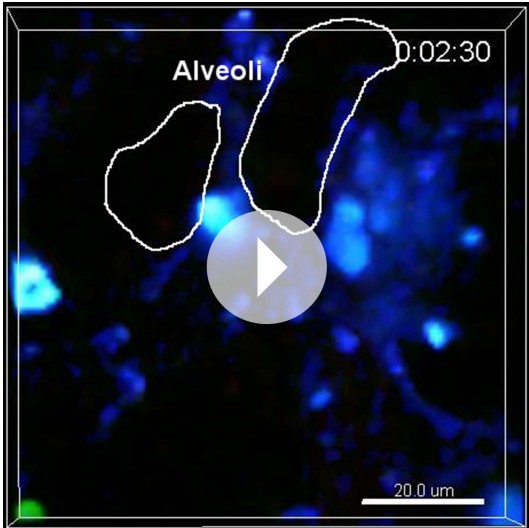

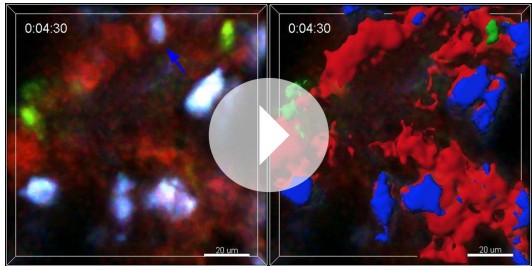

**Video 3.** Migratory behaviour of ECFP+ cells in lung vasculature. Live 3D imaging shows ECFP+ cells patrolling in the lumen of lung vessels (blue arrow) and interstitial ECFP+ cells in the vicinity of the vasculature. MacBlue×$Cx3cr1^{gfp/+}$ mice were injected with 2 MkDa rhodamine dextran 1 min before sacrifice. Lungs were imaged rapidly before leakage of the dye into the alveoli. The ECFP signal is in cyan, the EGFP signal in green, lung capillaries and vessels are in red, and the SHG signal is in blue.

**Video 2.** Migratory behaviour of lung interstitial ECFP+ cells. Live 3D imaging shows interstitial monocyte-derived cells scanning with protrusive activity or patrolling in the steady-state alveolar space of the lung of a MacBlue×$Cx3cr1^{gfp/+}$ mouse. The ECFP signal is in cyan, the EGFP signal in green, and the SHG signal (blue) indicates interstitial tissue, while white line drawings define the limits of alveoli.

*et al., 2013*). Lung mononuclear phagocyte subsets and neutrophils were labelled in different proportions (*Figure 9A*). AM were not labelled, confirming that the antibody did not reach the airways. Overall, 27% of interstitial macrophages, 12% of CD11b+ DC, and only 1% of CD11b− DC were labelled, suggesting a distinct location of these subsets related to the vasculature (*Figure 9A*). Up to 90% of Ly6C$^{high}$ and Ly6C$^{low}$ lung monocytes were labelled by the anti-CD45 antibody in vivo; however, the MFI was significantly lower compared to that of blood monocytes (*Figure 9B*). We excluded that this effect was due to collagen treatment, as similar results were obtained in non-digested lungs (*Figure 9B*). Furthermore, the phenotype of both Ly6C$^{high}$ and Ly6C$^{low}$ monocytes was slightly distinct from their circulating counterparts. Indeed lung monocytes displayed reduced expression of Ly6C, CD62L, and CD115 as previously observed, in addition to a higher level of CD11b, whereas only Ly6C$^{low}$ monocytes displayed higher CD11c expression (*Figure 9C*). The same differences were observed in monocytes isolated from lung without enzymatic digestion (data not shown). These results suggest that monocytes can be either extravascular but in close vicinity to the vessels or still intravascular and trapped in capillaries with reduced access to the bloodstream and submitted to the lung environment. MacBlue×$Cx3cr1^{gfp/+}$ mice were inoculated intravenously 5 min before sacrifice with a mixture of 10 μm and 0.2 μm red fluorescent beads to differentiate large vessel areas identified by the presence of both 10 μm and 0.2 μm beads, and lung capillary areas identified by only 0.2 μm beads (*Figure 9D*). In large vessels, ECFP+ cells were mainly round shaped, whereas in the vicinity of capillaries, interstitial ECFP+ cells displayed an elongated or amoeboid-like shape (*Figure 9D*). Confocal analysis of the microvascular area using CD31 staining showed that interstitial ECFP+ cells locate at the interface between the capillaries and the alveoli either intra- or extravascularly (*Figure 9E* and *Video 8*). To confirm that this positioning allows the sampling of particles inoculated by the airways route, we performed in vivo CD45 staining of monocytes that had captured beads 4 hr after intranasal inoculation (*Figure 9F*). CD45 in vivo labelling was exactly the same for both Ly6C$^{high}$ and Ly6C$^{low}$ monocytes that had captured the beads compared to non-phagocytic monocytes (*Figure 9F*). This result confirmed that lung monocytes are strategically positioned at the interface between the bloodstream and airways to survey both compartments.

## Discussion

*Jakubzick et al. (2013*) recently reported that monocytes enter and survey non-lymphoid organs (such as the lungs) at steady state. How this property differs from the already known functions of AM

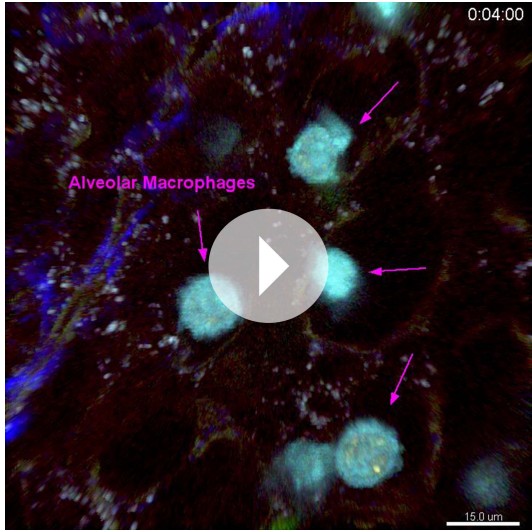

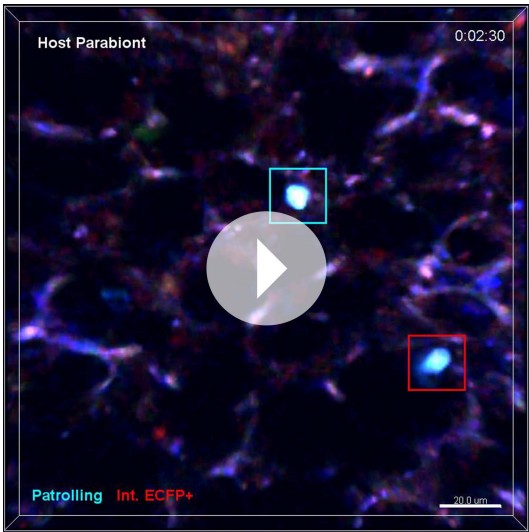

**Video 4.** Alveolar macrophage surveillance of the alveolar lumen. Live high-resolution 3D imaging of alveolar macrophages (indicated by pink arrows) in the alveoli of a MacBlue×*Cx3cr1*<sup>gfp/+</sup> mouse. The ECFP signal is in cyan, the EGFP signal in green, and the SHG signal in blue.

**Video 5.** Monocyte patrolling behaviour in host parabiont. Live 3D imaging shows interstitial monocyte patrolling behaviour (blue square) in the steady-state alveolar space of the lung of a C57Bl6 host parabiont with MacBlue×*Cx3cr1*<sup>gfp/+</sup> mouse 1 month after parabiosis.

and DC was still unknown. Fluorescent myeloid-specific reporter transgenic mice have been widely used to study the behaviour of myeloid cells in vivo, although no transgenic reporter is restricted to a particular myeloid lineage (*Hume, 2011*). LysM reporters can be used to track either neutrophils or monocytes (*Faust et al., 2000*). A wide diversity of cell subsets are labelled in the *Cx3cr1*<sup>gfp/+</sup> system, including NK cells, which can be excluded only with a specific γc-deficient background (*Auffray et al., 2007*). Similarly, the MacGreen (Csf1r-EGFP) transgene enables labelling of a wide range of mononuclear phagocytes as well as granulocytes (*Sasmono et al., 2003*). The MacBlue binary transgene, based upon a deleted Csf1r promoter, produced a transgenic line in which the reporter gene is extinguished in the majority of mature tissue macrophages and greatly reduced in granulocytes (*Ovchinnikov et al., 2008*; *Sauter et al., 2014*). Despite the fact that ECFP expression is not directly related to the level of Csf1r mRNA, it provides a unique marker of cells that have recently derived from monocytes.

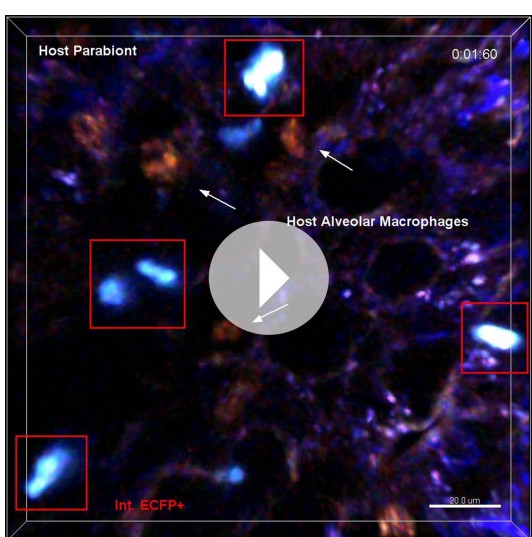

**Video 6.** Monocyte cell scanning behaviour in host parabiont. Live 3D imaging shows interstitial monocyte protrusive activity (red squares) in the steady-state alveolar space of the lung of a C57Bl6 host parabiont with MacBlue×*Cx3cr1*<sup>gfp/+</sup> mouse 1 month after parabiosis.

In the current study, we intercrossed *Cx3cr1*<sup>gfp/+</sup> and MacBlue mice to obtain an additional dimension of resolution and specificity. Monocytes were easily distinguishable from other subsets due to their common expression of ECFP. The differential expression of EGFP enabled a further distinction between Ly6C<sup>high</sup> and Ly6C<sup>low</sup> monocytes. Although our imaging approach did not allow us to distinguish between the two

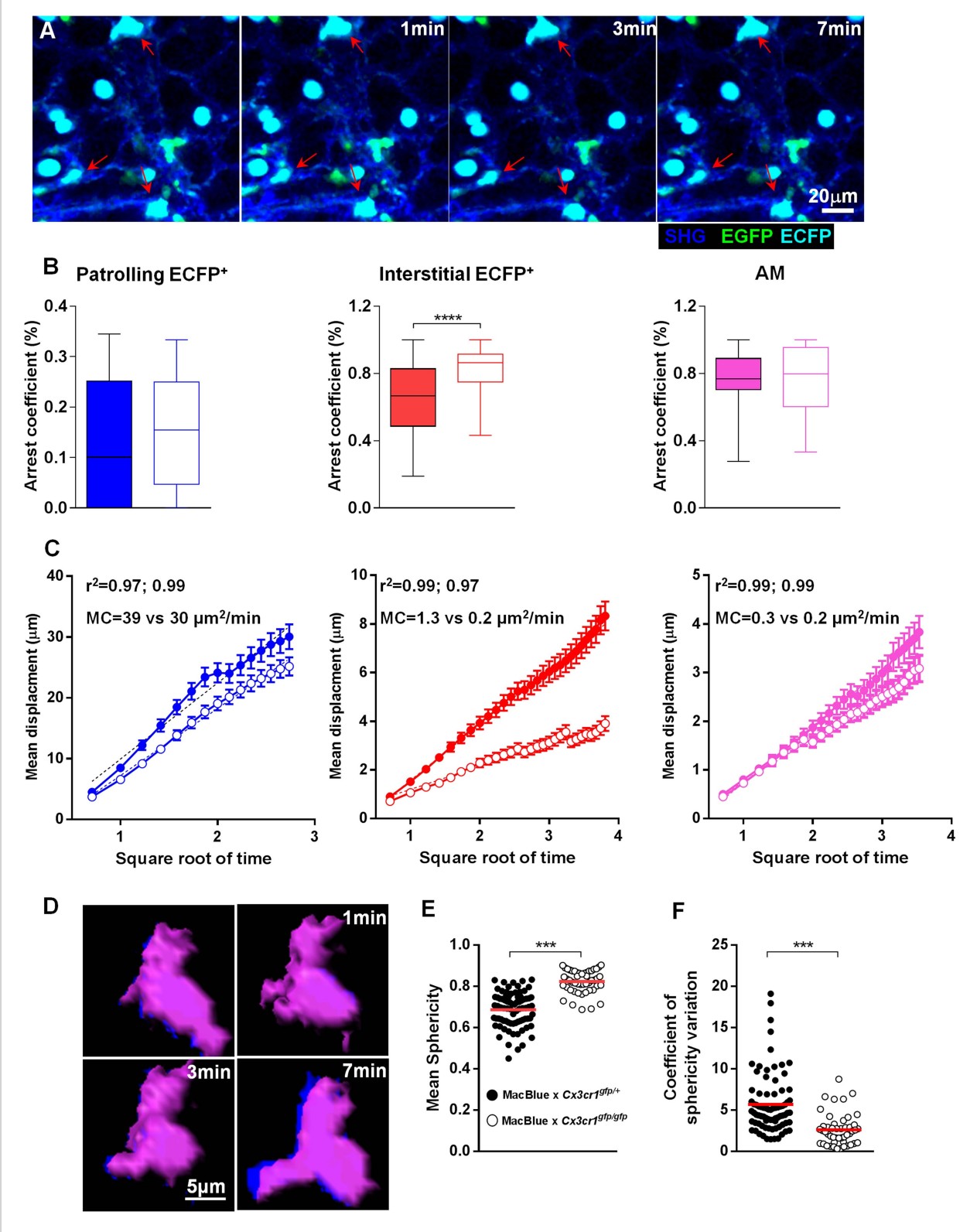

**Figure 6.** Interalveolar space scanning by monocyte-derived cells is CX3CR1 dependent. **(A)** Time series two-photon laser scanning microscopy (TPLSM) pictures show interstitial ECFP⁺ cell activity in explanted lungs of MacBlue×*Cx3cr1*^gfp/gfp^ mice. **(B)** Comparative box and whiskers analysis of the arrest coefficient of indicated cell subsets between MacBlue×*Cx3cr1*^gfp/+^ (full boxes) and MacBlue×*Cx3cr1*^gfp/gfp^ (empty boxes). Mann–Whitney tests were performed. Data are pooled from four to six mice from different experiments. **(C)** Comparison of the mean displacement ± SEM as a function of the

*Figure 6. continued on next page*

*Figure 6. Continued*

square root of time for patrolling interstitial ECFP[+] cells (blue), scanning interstitial ECFP[+] cells (red), and alveolar macrophages (AM) (pink) between MacBlue×*Cx3cr1*[gfp/+] (full circles) and MacBlue×*Cx3cr1*[gfp/gfp] (empty circles). Coloured dashed lines represent the linear regression of the curves. $r^2$ and motility coefficients (MC = x/4t) are indicated for the full versus empty circles, respectively. **(D)** Time series volume rendering image of interstitial ECFP[+] cell showing shape modifications. **(E)** Graph representing the mean sphericity of individual cells determined by measuring sphericity at each time point. Red bars represent the mean. **(F)** Graph representing the coefficient of sphericity variation for each cell tracked during 10 consecutive planes (5 min). Data are pooled from different movies from at least three different mice in each group from independent days. Student's t test was used.

monocyte subsets, the shape, size, tissue localization, and behaviour of the ECFP[+] cells provided a unique opportunity to track lung tissue mononuclear phagocytes and to compare their migratory patterns. The dual reporter also avoids potential artefacts created by the γc-deficient background excluding NK cells which we have shown to be highly represented in the CX3CR1-GFP[+] fraction within the lungs. The combination of CCR2 and CX3CR1 deficient mice and parabionts permitted a more precise characterization of the nature and origin of the different subsets and confirmed that interstitial ECFP[+] were monocyte-derived according to the proposed unified nomenclature (*Guilliams et al., 2014*).

Krummel's group previously performed functional imaging of lung DC using the *Itgax*-YFP reporter (*Thornton et al., 2012*). The combination of this reporter with the MacBlue mouse confirmed that interstitial monocyte-derived cells were distinct from lung DC. Histological analysis showed that monocyte-derived cells and DC share the surveillance activity through distinct regional distribution, the former located in the alveolar space at the interface with the bloodstream and the latter near larger bronchial airways. In vivo CD45 labelling added a new dimension of regional segregation between AM, interstitial macrophages, CD11b[+] DC, CD11b[−] DC, and lung monocytes. This specific regional segregation at steady state is likely the result of the rapid circulation of monocytes that arrest constitutively in the capillaries of the alveolar space with relatively short life span (*Jakubzick et al., 2013*), whereas DC are slow motile cells (*Thornton et al., 2012*). Of course regional segregation is likely less marked in the allergic asthma model for which important DC recruitment was observed in the vicinity of the alveoli, likely originating from monocytes (*Jakubzick et al., 2008*), where they exert longer transepithelial dendrites formation toward airways (*Thornton et al., 2012*). Our data demonstrate that the interstitial space of the lung is intensively surveyed by monocytes actively migrating or extending processes. The strategic positioning of lung monocytes at the interface allows for efficient capture of both blood and airway-derived particles. Four hours after intranasal inoculation, fluorescent particles remained undetectable in the mediastinal lymph nodes, as previously observed (*Jakubzick et al., 2008*; data not shown), however after 1 d, DC seem to prevail over monocytes in the transport of the inhaled particles. This suggests that the majority of monocytes may scavenge particles to clean up the tissue and filter the blood, whereas the majority of DC aim to transport antigen to the draining lymph node.

Due to the vascular permeability of the lung, it is challenging to address the blood/tissue partitioning of the subsets (*Jakubzick et al., 2013*). We combined in vivo CD45 labelling and confocal imaging to determine the precise location of lung monocytes (*Anderson et al., 2014*; *Girgis et al., 2014*). We determined that lung permeability does not allow it to be definitely concluded that lung

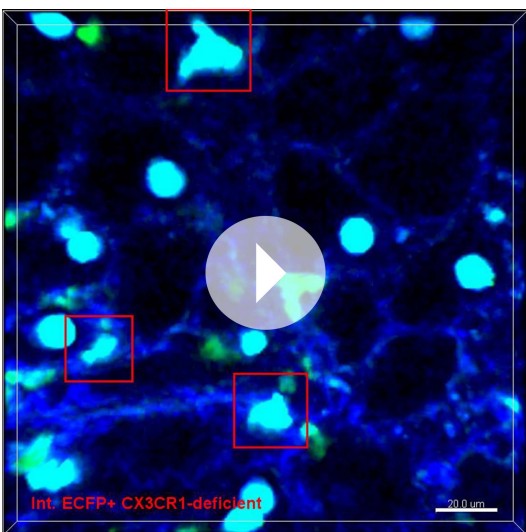

**Video 7.** Migratory behaviour of lung interstitial monocyte-derived cells in a MacBlue×*Cx3cr1*[gfp/gfp] mouse. Live 3D imaging shows reduced interstitial monocyte-derived cell protrusive activity (red square) in the steady-state alveolar space of the lung of a MacBlue×*Cx3cr1*[gfp/gfp] mouse. The ECFP signal is in cyan, the EGFP signal is in green, and the SHG signal (blue) indicates interstitial tissue and defines alveoli.

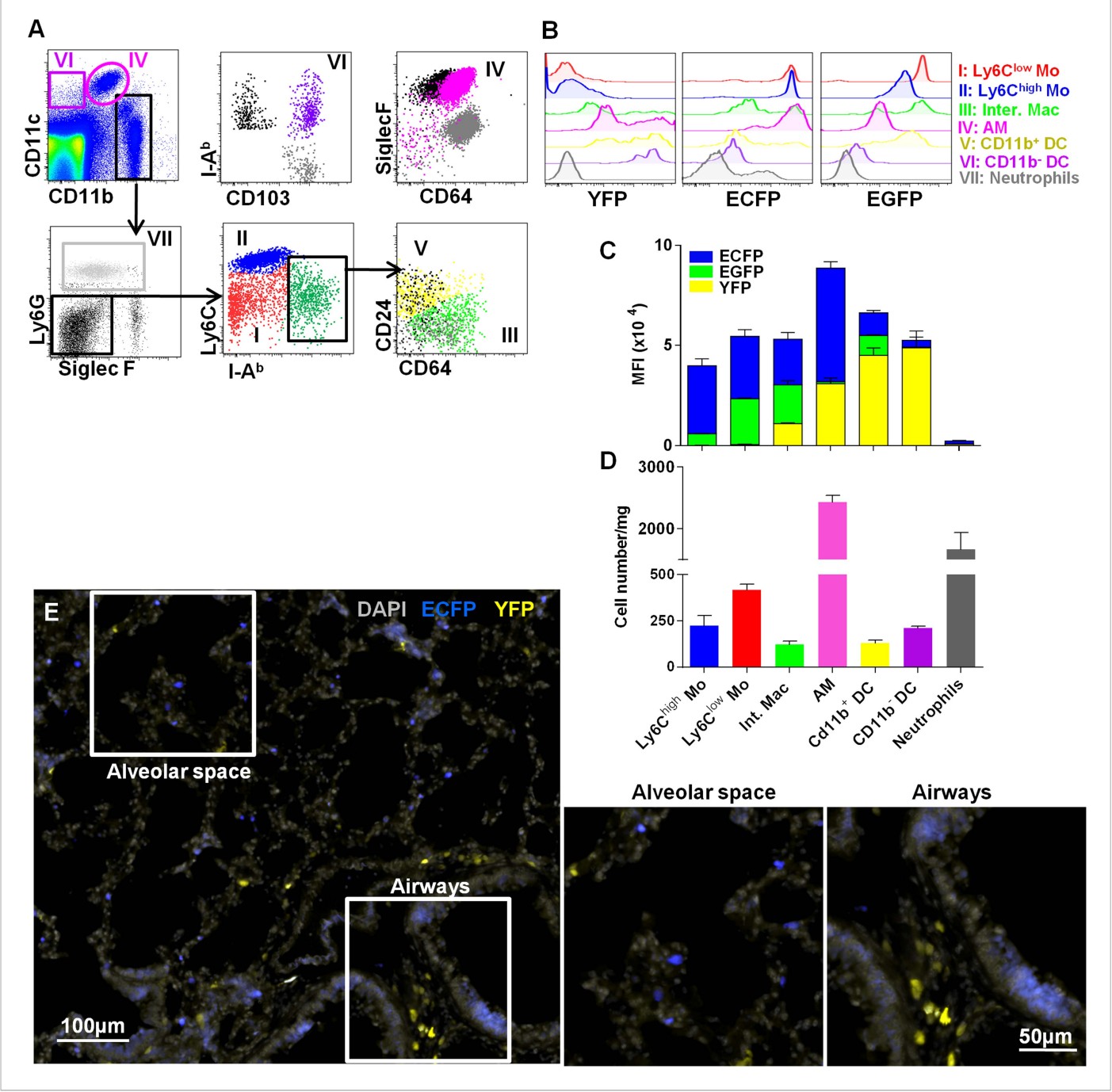

**Figure 7.** Interstitial monocyte-derived cells localized in the alveolar space whereas lung dendritic cells preferentially localized near large airways. **(A)** Dot plots show the gating strategy to define the different lung mononuclear phagocytes and dendritic cells from either MacBlue×*Cx3cr1^{gfp/+}* or MacBlue×*Itgax*-YFP mice. (I) Ly6C^{low} monocytes (red); (II) Ly6C^{high} monocytes (blue); (III) interstitial macrophages (light green); (IV) alveolar macrophages (AM) (pink); (V) CD11b^+ DC (yellow); (VI) CD11b^- DC (purple); and (VII) neutrophils (grey). For CD11b^- DC, AM, CD11b^+ DC, and interstitial macrophages (Inter. Mac), FMO (full minus one) signals gated on the specific subset were overlaid in black (for the x-axis) and grey (for the y axis). **(B)** Histogram plots represent the EGFP, ECFP, and YFP fluorescent reporter expression by each defined subset. YFP and EGFP were measured on individual mice. **(C)** Mean fluorescent intensity of EGFP (green), ECFP (blue), and YFP (yellow). Bars are mean ± SEM (n = 3 MacBlue×*Cx3cr1^{gfp/+}* and MacBlue×*Itgax*-YFP mice). The experiment has been repeated at least three times. **(D)** Absolute number of indicated subset per mg of lung (pooled data of n = 11 mice from at least three independent preparations). **(E)** Wide field image of MacBlue×*Itgax*-YFP mouse lung cryo-section showing ECFP^+ and YFP^+ cell distributions in alveolar space and near bronchial airways. Satellite images represent higher magnification of the corresponding white squares. Images are representative of three different mice. Mo, monocytes.

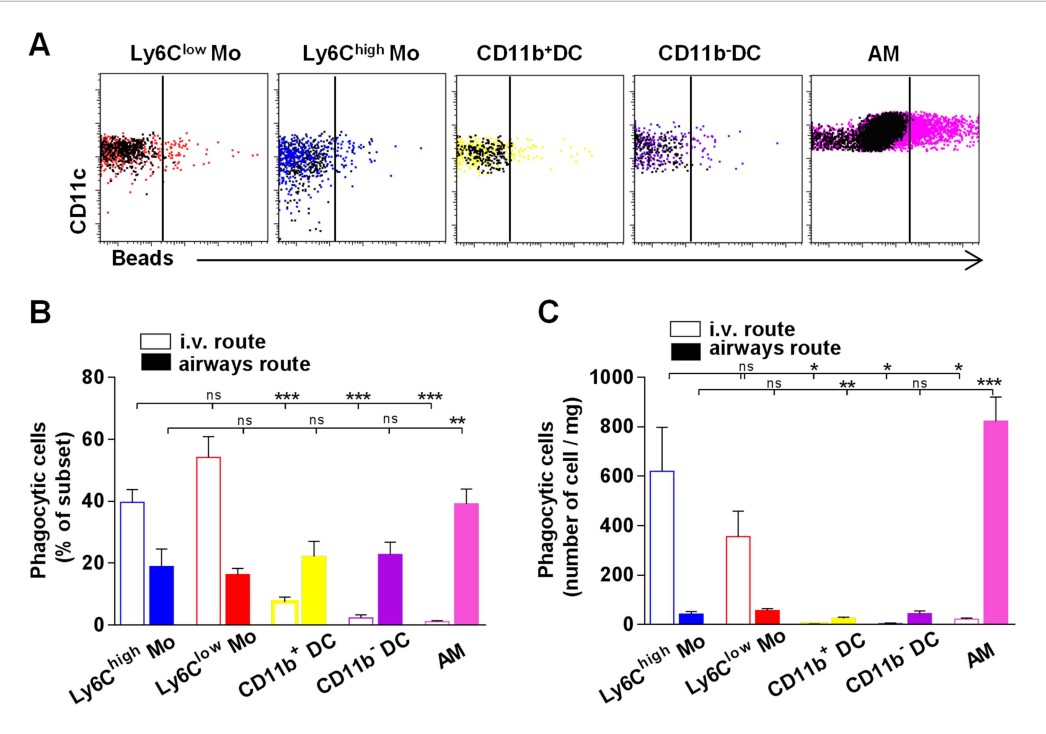

**Figure 8**. Lung monocyte-derived cells survey both airways and vascular routes whereas dendritic cells (DC) survey only airways. (A) Representative dot plots showing fluorescent bead uptake by lung mononuclear phagocytes and dendritic cells 4 hr after airway inoculation. Respective control subsets without beads are overlaid in black. (B) Graph represents the frequency of phagocytic cells as a percentage of the respective subset after intravenous injection (empty bars) or airway inhalation (full bars). (C) Graph represents the number of phagocytic cell subset per mg of collected tissue. Bars are mean ± SEM (n = 6–7 mice in each group from two to three different experiments). Student's t tests were performed to compare the phagocytic activity of all subsets to the referent population of Ly6C$^{low}$ monocytes. Mo, monocytes.

monocytes share their time either intra- or extravascularly or even both. However, we showed that lung monocytes have a slightly differentiated phenotype compared to the circulating monocytes with a downregulation of CD115 and CD62L and an upregulation of CD11b and CD11c. Secondly, the proportions of Ly6C$^{high}$ and Ly6C$^{low}$ monocytes were different in the lungs compared to the blood. Finally, monocyte localization in the vicinity of the bloodstream allows them to capture beads inoculated via the intranasal route.

Several studies used in situ imaging of the lung (*Kreisel et al., 2010*; *Looney et al., 2011*), but it is likely that working on lung explants may somehow affect the dynamics of the studied cells. For instance, fast circulating monocytes in the blood flow could not be observed. Nevertheless, we still observed high velocity patrolling cells that were present within large vessels, suggesting that this migration is independent of the blood flow. This approach provides at least the advantage of better stability without breathing constraints and avoids any accumulation of inflammatory monocytes due to inflammation. Other monocyte behaviour might be observed using in vivo imaging.

By contrast to the CX3CR1-dependence of endothelium patrolling (*Auffray et al., 2007*) as well as transepithelial dendrite formation in the gut (*Kim et al., 2011*), the patrolling activity of interstitial monocytes in the lung was unaffected by the absence of this receptor, whereas their scanning activity was severely impaired. Kim et al. showed that CX3CL1 was expressed by the epithelial cells of bronchioles and in the alveolar space using the CX3CL1-RFP reporters (*Kim et al., 2011*), arguing in favour of the role of this axis in protrusive activity of the CX3CR1$^+$ monocyte-derived cells. Because CX3CR1 is an important survival pathway, it is unclear whether these modifications are related to defects in chemotactic signal or cell survival. We did not observe defects in phagocytosis in CX3CR1-deficient mice (unpublished data), suggesting that the recently recruited monocytes are functional, arguing in favour of a role of CX3CR1 in their survival (*Kim et al., 2011*).

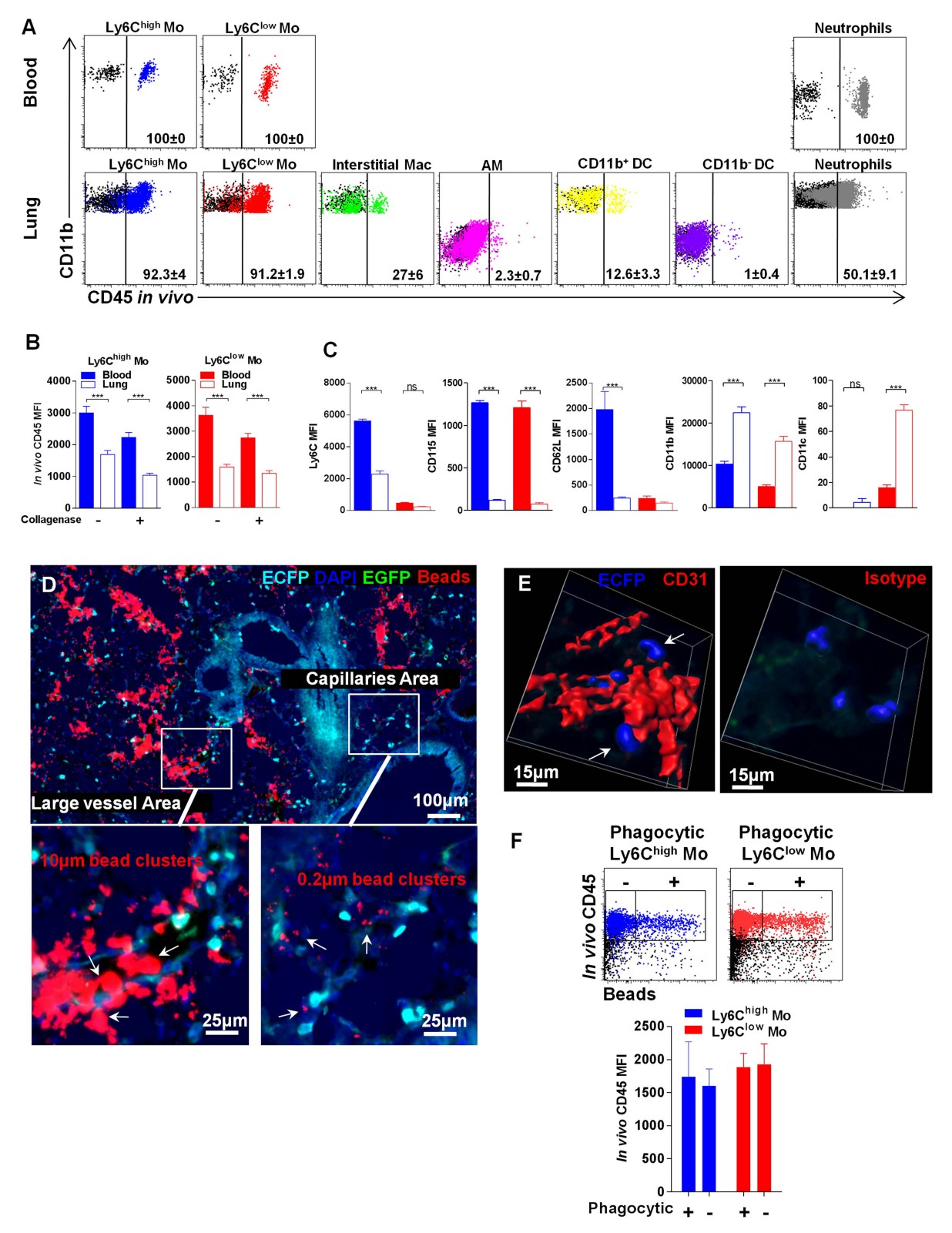

**Figure 9**. Lung monocyte-derived cells are located at the interface between blood and airways. (**A**) Representative overlayed dot plots of in vivo CD45 staining gated on blood and lung mononuclear phagocytes and neutrophils (coloured). CD45 staining control from mice not injected with anti-CD45 are shown (black). Percentage of CD45+ labelled cells according to control are indicated. Mac, macrophages. (**B**) Bars represent CD45 mean fluorescence intensity (MFI) after in vivo staining in Ly6Chigh (blue) and Ly6Clow (red) monocytes from the blood (full bars) and the lungs (empty bars) with or without

*Figure 9. continued on next page*

*Figure 9. Continued*

enzymatic digestion. **(C)** Bars represent MFI of the indicated markers gated on Ly6C[high] (blue) and Ly6C[low] (red) monocytes from the blood (full bars) and the lungs (empty bars). Bars represent mean ± SEM (n = 10 mice from two different experiments). ANOVA with Bonferroni adjustment was used. **(D)** MacBlue×*Cx3cr1*[gfp/+] mouse lung cryo-section showing ECFP[+] cell localisation in distinct vascular areas, according to bead distribution after intravenous injection of a mixture of 10 µm and 0.2 µm red fluorescent beads. Satellite images represent higher magnification of large vessels containing 10 µm and 0.2 µm beads, indicated by white arrows (left), and capillaries containing only 0.2 µm beads, indicated by white arrows (right). **(E)** Confocal volume rendering reconstitution image of CD31 (red) (left) or isotype staining (right) showing ECFP[+] monocytes (blue) in the vicinity of lung capillaries. Volume rendering reconstruction has been determined according to the isotype staining. **(F)** Dot plots represent in vivo CD45 staining of Ly6C[high] (blue) and Ly6C[low] (red) monocytes 4 hr after intranasal inoculation of fluorescent beads. CD45 staining control from mice not injected with anti-CD45 is shown (black). Bars represent the CD45 MFI gated on phagocytic (beads[+]) and non-phagocytic (beads[−]) monocytes. Bars represent mean ± SEM (n = 4 mice from two different experiments).

Overall, our study provided important fundamental insights into the behaviour of tissue monocytes during the process of immune surveillance in comparison to resident macrophages and DC in a key biological tissue. We concluded that tissue monocytes represent a major first line phagocytic compartment in comparison to DC. Monocyte-derived cells developed an organized distribution at the interface between blood and airways with a specific pattern of movements in the lungs to enable them to rapidly detect danger, trigger inflammation, capture antigen, and undergo subsequent immune response. Understanding these activities is a key step towards improving the treatment of a wide range of inflammatory diseases and also vaccination strategies targeting the route of antigen uptake.

## Materials and methods

### Mice

*Cx3cr1*-GFP-Kin (*Cx3cr1*[gfp/+]) or *Itgax*-YFP transgenic mice (CD11c-YFP) (*Lindquist et al., 2004*) and *Csf1r*-Gal4VP16/UAS-ECFP (MacBlue) (*Ovchinnikov et al., 2008*) were intercrossed to generate *Cx3cr1*[gfp/+] ×*Csf1r*-Gal4VP16/UAS-ECFP mice herein called MacBlue×*Cx3cr1*[gfp/+] or MacBlue×*Cx3cr1*[gfp/gfp], and MacBlue×*Itgax*-YFP, respectively. These new strains were bred in the Nouvelle Animalerie Commune animal facility at Pitié-Salpêtrière. *Cx3cr1*[gfp/+]-*Ccr2*[rfp/+] mice were kindly provided by Israel Charo (Gladstone Institute, San Francisco, CA, USA) (*Saederup et al., 2010*) to generate *Cx3cr1*[gfp/gfp]-*Ccr2*[rfp/+], *Cx3cr1*[gfp/+]-*Ccr2*[rfp/-], and *Cx3cr1*[gfp/gfp]-*Ccr2*[rfp/rfp] mouse strains. All mice were used between 8 and 12 weeks of age. C57Bl6 female host parabionts were generated with MacBlue×*Cx3cr1*[gfp/+] females and analysed after 1 month of parabiosis. Blood leukocyte chimerism was evaluated 2 weeks after surgery, and showed a T cell chimerism of 50 ± 10%. In these settings, monocyte chimerism from B6 to MacBlue×*Cx3cr1*[gfp/+] was around 37 ± 14% and MacBlue×*Cx3cr1*[gfp/+] to B6 was reduced to 14 ± 13%.

All experiment protocols were approved by the French animal experimentation and ethics committee and validated by Service Protection et Santé Animales, Environnement (no. A-75-2065). Sample sizes were chosen to ensure the reproducibility of the experiments and according to the 3Rs of animal ethics regulation.

### In vivo phagocytosis and CD45 labelling

A total of 10[11] FluoroSpheres carboxylate–modified microspheres (200 nm) (Invitrogen, Eugene, OR, USA) were inoculated either by intravenous or airway routes. Phagocytosis by lung subsets was analysed 4 hr after inoculation by flow cytometry. Bead inhalation was performed by loading a 10 µl drop of NaCl 0.9%

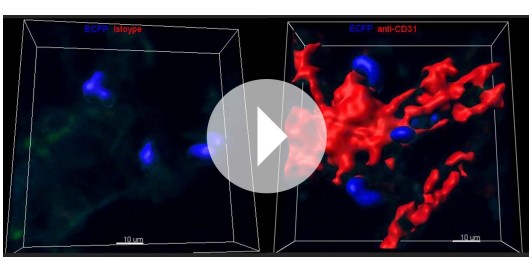

**Video 8.** Confocal 3D reconstruction of lung ECFP[+] cells in the lung vasculature. High resolution 3D reconstruction shows lung ECFP[+] cell localisation at the interface between lung capillaries and airways. Lung vasculature has been stained with anti-CD31 (right) or isotype control (left) in a lung cryo-section of a MacBlue×*Cx3cr1*[gfp/+] mouse. Acquisition parameters were identical for both stainings. Volume rendering parameters were determined based on CD31 isotype staining. Capillaries are in red and ECFP[+] cells are in blue.

bead solution in each nostril of mice anaesthetized by an intraperitoneal injection of a mixture of ketamine/xylazine (100 and 10 mg/kg body weight, respectively). Mice for which bead inhalation failed have been excluded from the analysis.

For in vivo CD45 labelling, mice were injected intravenously with 1 µg of anti-CD45 (clone 30-F11). Mice were sacrificed 2 min after blood was drawn. Lungs were harvested and bathed in a large volume of PBS to dilute free antibody.

## Flow cytometry

Flow cytometry was performed with the flow cytometer FACScanto or Fortessa (BD, Franklin Lakes, NJ, USA) and DIVA Flow Cytometry acquisition software, and was analysed with FlowJo software (Tree Star, Ashland, OR, USA). After blood was drawn via retro-orbital puncture with heparin, the mouse lung vasculature was gently flushed with an intracardiac injection of PBS until complete blood clearance. For in vivo CD45 staining experiments, the lung vasculature was not flushed. Lungs were then harvested and digested in RPMI medium (Gibco, Invitrogen, Cergy Pontoise, France) with 1 mg/ml collagenase IV (Sigma, St Quentin Fallavier, France) for 30 min at 37°C and/or directly (without collagenase incubation) dissociated through a 70 µm-pore cell strainer (Becton Dickinson, Rungis, France) to obtain the cell suspension. Similarly, for some experiments blood was or was not incubated with 1 mg/ml collagenase IV. For antibody staining, 50 µL of cell suspension was incubated with 1 µg/mL purified anti-CD16/32 (2.4G2) (BD Biosciences, San Jose, CA, USA) for 10 min at 4°C and for an additional 20 min with the appropriate dilution of specific antibodies. The panel of antibodies used was: anti-CD11b (clone M1/70), anti-Ly6C (clone AL-21), anti-Ly6G (clone 1A8), anti-NK1.1 (clone PK136 ), anti-CD45 (clone 30-F11), anti-CD11c (clone HL3), anti-I-A$^b$ (clone AF6-120-1), anti-CD 62L (clone MEL-14), anti-SiglecF (clone E50-2440), anti-CD24 (cloneM1/69), anti-CD103 clone (M290), rat IgG2b isotype control (BD Biosciences), F4/80 (clone BM8), CD115 (clone AFS98), rat IgG2a isotype control (clone eBr2a; eBioscience, San Diego, CA, USA), and anti-CD64 (clone X54-5/7.1.1) (BioLegend, San Diego, CA, USA). After incubation, cell suspensions were washed once in 0.5% BSA/2 mM EDTA in PBS and analysed directly by flow cytometry. For blood samples, erythrocytes were lysed with a lysis buffer containing 0.15 M $NH_4Cl$, 0.01 mM $KHCO_3$ and 0.1 mM EDTA and resuspended in 0.5% BSA/2 mM EDTA in PBS. FMO staining controls (full minus one) have been performed for all sets of experiment and are indicated in dot plots or histogram plots when necessary. Specific FMO gating for alveolar macrophages was required due to their bright autofluorescence. Absolute numbers were calculated by adding to each vial a fixed number (10,000) of non-fluorescent 10-µm Polybead Carboxylate Microspheres (Polysciences, Niles, IL, USA) according to the formula: no. cells = (no. acquired cells × 10,000)/(no. acquired beads).

## Tissue processing for histology analysis

Briefly, organs were harvested and fixed in 10% formalin for 4 hr then incubated in 30% sucrose-PBS overnight at 4°C before being embedded in Tissue–Tek OCT compound (Sakura Finetek, Alphen aan den Rijn, Netherlands) and frozen at −80°C. Sectioning was completed on a HM550 Cryostat (Thermo Fisher Scientific, Waltham, MA, USA) at −20°C and 5 µm sections were collected on Superfrost Plus Slides (Thermo Fisher Scientific) and stored at −20°C until use.

Vascular staining using beads was performed on histological sections of lung from MacBlue×Cx3cr1$^{gfp/+}$ mice injected 5 min before sacrifice with a mixture of 10 µm Polybead Carboxylate Microspheres and 0.2 µm FluoroSpheres carboxylate. Lung sections were rehydrated with PBS for 10 min, counterstained and mounted with Vectashield Mounting Medium with DAPI (4,6-diamidino-2-phenylindole; Vector Laboratories). Imaging used a Zeiss Axio Microscope (Carl Zeiss, Oberkochen, Germany). CD31 vascular staining was performed on lung cryo-sections from MacBlue×Cx3cr1$^{gfp/+}$ mice. A first blocking step was performed with 3% BSA/PBS solution for 30 min. Slides were then incubated for 1 hr at 37°C with the primary antibody rat anti-mouse CD31 (4 µg/ml) (clone MEC 13.3; Becton Dickinson, San Jose, CA, USA) or the isotype control (4 µg/ml) (clone eBR2a; eBioscience). Slides were next incubated with Avidin/Biotin Blocking Kit (SP-2001; Vector Laboratories, Burlingame, CA, USA) and then stained with a biotinylated donkey anti-rat IgG at 3 µg/ml for 30 min at room temperature. After three (5 min) washes in PBS, slides were incubated with Cy3-conjugated streptavidin at 2.6 µg/ml for 30 min at room temperature (Jackson ImmunoResearch Laboratories, West Grove, PA, USA). Slides were counterstained and mounted with Vectashield

Mounting Medium with DAPI and analysed by using a Zeiss LSM 710 NLO confocal microscope coupled with 458 nm, 488 nm, and 543 nm lasers to detect ECFP, EGFP, and Cy3 simultaneously on three photomultipliers.

Acquisition settings were identical for both isotype and CD31 staining. Volume rendering was performed using Imaris software (Bitplane, Zurich, Switzerland) and parameters were set according to CD31 isotype staining.

## Multiphoton imaging

Freshly explanted lungs were immobilized in an imaging chamber perfused with oxygenated (95% $O_2$ plus 5% $CO_2$) RPMI medium containing 10% FCS. The local temperature was monitored and maintained at 37°C. For some experiments, 2 MkDa rhodamine dextran was injected intravenously 1 min before euthanasia and lung vasculature was ligatured to reduce leakage of the dye. The two-photon laser scanning microscopy (TPLSM) set-up used consisted of a Zeiss LSM 710 NLO multiphoton microscope (Carl Zeiss) coupled to a Ti:Sapphire crystal laser (Coherent Chameleon Ultra, Santa Clara, CA, USA), which provides 140 fs pulses of NIR light, selectively tunable between 680 and 1080 nm, and an acousto-optic modulator to control laser power. The system included three external non-descanned detectors that enabled the simultaneous recording of three fluorescent channels with a combination of two dichroic mirrors (565 nm and 690 nm), 565/610 and 500/550 bandpass filters, and a 485 lowpass filter. The excitation wavelength was 870 nm. Cell motility was measured every 30 s by five consecutive 3 μm z spacing stacks (total thickness of 12 μm) with a plan apochromat × 20 (NA = 1) water immersion objective.

Fluorescent cells were monitored over time with three-dimensional automatic tracking and manual correction with Imaris software (Bitplane). The different cell subsets were defined according to the fluorescent signature, the size, the shape, the localization, and the behaviour. Typically AM are ECFP[bright], large, round, and sessile cells located in the lumen of alveoli. Patrolling monocytes are ECFP+ and small with an amoeboid-like shape displaying strong displacements in the interalveolar space. Interstitial monocyte-derived cells are ECFP+ sessile cells displaying protrusive activity in the interalveolar space. Interstitial EGFP+ are either small and round (likely NK) in the lung alveolar space, or dendritic-shaped in the pleura and along airways. Cells that could not be tracked for more than 2 min were not considered. The arrest coefficient was defined as the proportion of time each cell's instantaneous velocity (calculated for every 30 s interval) was less than 2 μm/min. The MC was determined on a 2D-based analysis by z projection of the 3D stacks, using the formula (x/4 t) (where x represents the slope of the mean displacement as a function of the square root of time) (*Cahalan and Parker, 2008*) (see statistical section). Coefficient of sphericity variation was determined by calculating the coefficient of variation with Graphpad Prism (Graphpad, San Diego, CA, USA) of the sphericity determined for each cell on 10 consecutive planes (5 min). Velocity and sphericity were determined using Imaris (Bitplane). The acquisition and analysis protocols for all experimental conditions to be compared were identical.

## Statistical analysis

All statistical analyses were performed with Graphpad Prism. For intravital analysis of cell behaviour, each sample value was first tested for Gaussian distribution by the D'Agostino and Pearson omnibus normality test. Accordingly, multigroup comparison tests were performed by one-way ANOVA for parametric distribution followed by Bonferroni multiple comparison test or Kruskal–Wallis test for non-parametric distributions, followed by Dunn's multiple comparison test. For simple comparison analysis, Student's t test was performed to compare parametric distribution and Mann–Whitney rank sum tests were performed to compare non-parametric distribution. For MC measurement, the slope of the mean displacement as a function of the square root of time (*Cahalan and Parker, 2008*) was calculated by linear regression statistical analysis. Linearity was considered significant for r >0.9. Symbols used: *, p<0.05; **, p<0.01; ***, p<0.001.

## Acknowledgements

The authors wish to thank Jo Ann Cahn for manuscript editing, the Plateforme Imagerie Pitié-Salpêtrière (PICPS) for assistance with the two-photon microscopes, and the animal facility 'NAC' and

Camille Baudesson for mice breeding assistance. The research leading to these results has received funding from the European Community's Seventh Framework Programme (FP7/2007–2013) under grant agreement no. 304810 – RAIDs and no. 241440 – Endostem, from Inserm, from Université Pierre et Marie Curie 'Emergence', from 'La Ligue contre le cancer', from 'Association pour la Recherche sur le Cancer', and from 'Agence Nationale de la Recherche' Programme Emergence 2012 (ANR-EMMA-050). PH was supported by 'La Ligue contre le cancer'.

The authors declare no competing financial interests.

## Additional information

### Funding

| Funder | Grant reference | Author |
| --- | --- | --- |
| European Commission (EC) | n 304810 | Alexandre Boissonnas |
| European Commission (EC) | n 241440 | Christophe Combadière |
| Ligue Contre le Cancer | | Alexandre Boissonnas |
| Association pour la Recherche sur le Cancer | | Alexandre Boissonnas |
| Agence Nationale de la Recherche | Emergence 2012 (ANR-EMMA-050) | Christophe Combadière |

The funders had no role in study design, data collection and interpretation, or the decision to submit the work for publication.

### Author contributions

MPR, LP, P-LL, Conception and design, Acquisition of data, Analysis and interpretation of data, Drafting or revising the article; PH, FL, CP, Acquisition of data, Drafting or revising the article; DAH, Drafting or revising the article, Contributed unpublished essential data or reagents; CC, Conception and design, Drafting or revising the article, Contributed unpublished essential data or reagents; AB, Conception and design, Acquisition of data, Analysis and interpretation of data, Drafting or revising the article, Contributed unpublished essential data or reagents

### Ethics

Animal experimentation: All experiment protocols were approved by the French animal experimentation and ethics committee and validated by "Service Protection et Santé Animales, Environnement" (no. A-75-2065). Sample sizes were chosen to ensure reproducibility of the experiments and according to the 3Rs of animal ethics regulation.

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
