## [Decision Letter]

Thank you for submitting your work entitled "Immune surveillance of the lung by migrating tissue-monocytes" for peer review at *eLife*. Your submission has been favorably evaluated by Tadatsugu Taniguchi (Senior editor), a Reviewing editor, and three reviewers. The following individuals responsible for the peer review of your submission have agreed to reveal their identity: Ronald Germain (Reviewing editor); Steffen Jung, Miriam Merad, and Gwendalyn Randolph (peer reviewers).

The reviewers have discussed the reviews with one another and the Reviewing editor has drafted this decision to help you prepare a revised submission.

Summary::

This manuscript investigates the behavior of recently recruited monocytes in mouse lung. Monocytes are leukocytes classically considered to be located in the blood and believed to differentiate upon extravasation into macrophage- or DC-like cells. Recently though, a tissue-resident monocyte population has been reported. The exact dwell times of these cells in the tissue, their fates and most importantly their functional relevance remain unclear. The study of blood monocytes and tissue monocytes, and the definition of their functions, is among most important outstanding issues in the field. At the same time, it is highly challenging due to the dynamics of the system. The major novelty of the work, and a development that will be of broad interest to readers beyond those in pulmonary immunology, is the development of a mouse strain that displays unique combinations of single or double fluorescent tags, allowing one to separately distinguish monocytes from bona fide macrophages or dendritic cells. The authors employ these new tagged animals to investigate the concept previously published by Jakubzick et al., Immunity 2013 that monocytes in a relatively undifferentiated state inhabit the parenchyma of the lung and display unique functions/behavior compared with macrophages or DCs. The authors carry out 2-photon microscopy to show that monocytes and DCs are highly motile in the lung but localize to distinct anatomical regions. This is an exciting finding since it suggests that both cell types might survey the organ in distinct spaces and, as shown by Jakubzick et al., transport antigen from these different locations to the draining LNs.

A large portion of the manuscript is rightly devoted to proving that *Cx3cr1-GFP*/ *Csf1r-EGFP* double reporter positive cells are monocyte-derived cells. The authors then provide data on the migration properties of the cells and perform functional studies by challenging the mice intravenously or intra-tracheally with fluorescently-labeled latex beads and monitoring phagocytosis by the different mononuclear phagocytes. They provide evidence that tissue monocytes survey both the vascular and alveolar compartments, while surveillance of tissue resident DC and macrophages is restricted to material delivered by the tracheal route.

Altogether, this is an important study that introduces a novel, though complex, experimental system to identify tissue monocytes. The authors provide some initial functional characterization of the cells, although the latter could clearly be substantiated, for instance by the addition of a pathogen challenge.

Essential revisions:

1) The authors argue that the labeled cells are in tissue because they performed an extensive perfusion and the cells in lung differ in phenotype and subset distribution from the cells found in blood. However, to strengthen this critical point they should determine blood/tissue partitioning, using in vivo CD45 labeling, for instance as recently reported by Girgis et al. (PLOS Pathogen 2014), at least in their flow cytometric analysis but preferentially also in the imaging.

2) Among lung macrophages, alveolar macrophages are the least dependent on Csf1 since they depend on Csf2 for their survival and maturation locally, and therefore it is unclear why they should express higher levels of the Csf1R? Can the authors clarify whey they express higher ECFP levels than interstitial macrophages.

3) The demonstration that monocytes do extravasate into the interstitial space needs to be further established using confocal analysis of lung sections stained with different endothelial markers.

4) Why do monocytes fail to differentiate into macrophages in the non-inflamed lung? Would they convert if macrophages are ablated in the absence of overt inflammation? While this may be experimentally difficult to address in the near term, there should be some discussion of this issue.

5) The authors have chosen to carry out their imaging using lung explants. In situ imaging of the lung using 2-photon microscopy has been reported by several groups (Kreisel, D. et al., PNAS, 2010; Looney MR, Thornton EE, Sen D, Lamm WJ, Glenny RW, Krummel MF. Nat Methods. 2011 January;8(1):91-6). There can be differences in behavior using explants vs direct intravital imaging and in the absence of additional studies using direct in vivo methods, the authors should be careful in their interpretations of dynamics.

6) Like Jakubzick et al., the authors make use of intranasally instilled beads as phagocytic cargo for DCs, monocytes, etc. The authors conclude that monocytes are more proficient at clearing cargo than DCs. But it is important to point out that it depends upon what one means by clearance. The authors seem to mean the quantity of material engulfed by monocytes and in that sense it appears that macrophages were the most proficient. However, Jakubzick et al. showed that DCs were still far more likely than monocytes to "clear" cargo by migrating with it to the draining lymph node, although monocytes could do this too. So when it comes to moving cargo to the LN, DCs seem to prevail. It does not appear that the authors separately investigated this issue, but it would be important to clarify the meaning of "clearance" and make sure that they do not inadvertently imply migratory clearance. Better citation of past studies that used particulate antigen tracers would be desirable in general.

Minor points:

1) Both Ly6C^high^ and Ly6C^low^ monocytes express surface CX3CR1, as shown by Geissmann et al. The authors should state this more clearly in their Introduction.

2) Parabiosis: To ensure that parabiosis led to efficient mixing in the parabionts, the authors should provide the number of mixed circulating T cells, B cells, neutrophils and monocytes in parabionts.

3) The movies could be rendered more aesthetically pleasing and easier to follow if more time was spent in processing them for display to readers.

4) It might be worthwhile to indicate in Figure 2 that the difference in CD115 staining is likely due to cleavage in response to digestive enzymes used.

5) The number of alveolar macrophages, interstitial macrophages and DC in the lung should also be quantified.

---

## [Author Response]

*1) The authors argue that the labeled cells are in tissue because they performed an extensive perfusion and the cells in lung differ in phenotype and subset distribution from the cells found in blood. However, to strengthen this critical point they should determine blood/tissue partitioning, using* in vivo *CD45 labeling, for instance as recently reported by Girgis et al. (PLOS Pathogen 2014), at least in their flow cytometric analysis but preferentially also in the imaging*.

Interpretation of in vivo CD45 labelling has been shown to be difficult in the lung due to the vascular permeability. Jakubzick et al. 2013 state: “The method was not useful in lung; resident DCs were labeled as were lung monocytes (Figure 4), indicating that anti- CD45 mAb readily accessed the highly vascularized lung.”

We performed this experiment by injecting i.v. 1 µg of anti-CD45, 2 min before sacrifice and showed that lung mononuclear phagocyte subsets and neutrophils were labelled in different proportions (See new Figure 9). Up to 90% of Ly6C^high^ and Ly6C^low^ lung monocytes were labelled but the mean fluorescence intensity (MFI) for CD45 was significantly lower than that of blood monocytes suggesting a free access to CD45 staining but reduced compared to as mentioned above. We excluded that this effect was due to collagenase treatment, as similar results were obtained in non-digested lungs (See Figure 9).

We analysed the MFI of different markers expressed by lung monocytes labelled with anti-CD45 and confirmed that lung monocytes showed downregulation of the expression of Ly6C, CD115 and CD62L and upregulation of the expression of integrins CD11b and CD11c compared to blood monocytes (Figure 9). Again, this was not due to collagenase treatment, as similar results were obtained in non-digested blood and lungs. See Figure 10. Collagenase digestion is however necessary to isolate monocytes from the lung tissue (right panel).

Author response image 1.(Left colored panels) MFI of indicated markers expressed by Ly6C^high^ Mo (blue) and Ly6C^low^ MP (red) of the blood (full bar) and the lung (empty bars). Cell preparations were performed in the absence of enzymatic digestion for both blood and lungs. (Right black and white) Bar indicated the number of lung monocytes collected with or without collagenase digestion. (n = 8–10 mice from 2 independent experiments, ANOVA test with Bonferoni adjustment was performed ***, p < 0.001).**DOI:**
http://dx.doi.org/10.7554/eLife.07847.023

These observations suggest that lung monocytes are in close contact with the lung vasculature, either hanging on the luminal side of the vessel or spreading on the tissue side but have already undergone a process of differentiation and are thus distinct from circulating monocytes.

After i.v. inoculation of a mixture of 10 µm and 0.2 µm fluorescent beads, we observed that the lung vasculature is organized in large vascular areas stained by both bead sizes and capillary areas only labelled by 0.2 µm beads. Lung monocytes localized in both areas (Figure 9). Confocal imaging after CD31 staining showed that monocyte located in close contact with capillaries either inside or at the interface between CD31 + cells and lung tissue (Figure 9).

Finally we performed in vivo CD45 labelling of lung monocytes after intranasal inoculation of beads.

We show that the intensity of CD45 labelling was identical in phagocytic and non-phagocytic monocytes (Figure 9). This result firmly confirmed that lung monocytes localisation, either intra or extravascular, provides the ability to capture beads in the airways.

These observations are in accordance with the difference observed between lung monocytes and lung DC in their ability to survey both blood vasculature and airways. We have added a new Figure 9 and discussed the results in the manuscript (subsection “Lung monocyte-derived cells are located at the interface between blood and airways”.)

*2) Among lung macrophages, alveolar macrophages are the least dependent on Csf1 since they depend on Csf2 for their survival and maturation locally, and therefore it is unclear why they should express higher levels of the Csf1R? Can the authors clarify whey they express higher ECFP levels than interstitial macrophages*.

In the MacBlue mice, the expression of the ECFP transgene is not directly related to the level of Csf1r mRNA. The promoter used has a deletion of the upstream promoter element that seems to be required for expression in most tissue macrophages (Sauter et al., PLosOne 2014). Alveolar macrophages, along with microglia and Langerhans cells, do not seem to require this element and so are uniquely labelled with ECFP in the MacBlue line. We commented this point (subsection “Lung monocyte-derived cells are located at the interface between blood and airways”).

As underlined by the reviewers, Schneider et al. recently describe the mandatory role of CSF2 in alveolar macrophages maturation (Schneider et al., 2014). However Schulz et al. demonstrate using a fate mapping approach that alveolar macrophages arise from the maturation of a non-hematopoietic Csf1r + precursor (Schulz et al., Science 2012). High expression of the Csf1R transgene by the alveolar macrophages has also been previously described in the Macgreen mice. In addition, treatment with a CSF1R antibody results in alveolar macrophages depletion (Mc Donald et al., blood 2010). Thus alveolar macrophages also depend on CSF1. The relative contribution of CSF1 and CSF2 in maturation and survival of alveolar macrophages deserve further investigation.

*3) The demonstration that monocytes do extravasate into the interstitial space needs to be further established using confocal analysis of lung sections stained with different endothelial markers*.

Confocal analysis of lung sections stained with anti-CD31 and intravital imaging of lung explants after rhodamine dextran injection have been performed (see comments above). In addition with in vivo CD45 labelling experiments, we conclude that lung monocytes localize at the interface of the vasculature and the airways either inside or outside the capillaries, giving the ability to survey both routes (Figure 9 and Videos 3, 4, 5, 6, 7 and 8).

*4) Why do monocytes fail to differentiate into macrophages in the non-inflamed lung? Would they convert if macrophages are ablated in the absence of overt inflammation? While this may be experimentally difficult to address in the near term, there should be some discussion of this issue*.

In fact we do see a minimal differentiation state of the lung monocytes: down regulation of Ly6C CD62L, CD115 and upregulation of CD11b and CD11c.

Considering the important proportion of lung monocytes compared to interstitial macrophages and DCs (see new quantification in Figure 9) we suspect that the life span of lung monocytes or the time of residency may be too short to fully differentiate into macrophages. However, we do not exclude that a proportion of interstitial macrophages that express ECFP and EGFP like monocytes and higher expression of class II may origin from monocytes.

These results are now presented and discussed in the manuscript.

*5) The authors have chosen to carry out their imaging using lung explants.* In situ *imaging of the lung using 2-photon microscopy has been reported by several groups (Kreisel, D. et al., PNAS, 2010; Looney MR, Thornton EE, Sen D, Lamm WJ, Glenny RW, Krummel MF. Nat Methods. 2011 January;8(1):91-6). There* can *be differences in behavior using explants vs. direct intravital imaging and in the absence of additional studies using direct* in vivo *methods, the authors should be careful in their interpretations of dynamics*.

We agree with this comment and discussed this point in the manuscript (Discussion).

*6) Like Jakubzick et al., the authors make use of intranasally instilled beads as phagocytic cargo for DCs, monocytes, etc. The authors conclude that monocytes are more proficient at clearing cargo than DCs. But it is important to point out that it depends upon what one means by clearance. The authors seem to mean the quantity of material engulfed by monocytes and in that sense it appears that macrophages were the most proficient. However, Jakubzick et al. showed that DCs were still far more likely than monocytes to "clear" cargo by migrating with it to the draining lymph node, although monocytes could do this too. So when it comes to moving cargo to the LN, DCs seem to prevail. It does not appear that the authors separately investigated this issue, but it would be important to clarify the meaning of "clearance" and make sure that they do not inadvertently imply migratory clearance. Better citation of past studies that used particulate antigen tracers would be desirable in general*.

We agree that the interpretation could be confusing. In our experiment we only consider clearance through beads capture. Four hours after intranasal injection we were not able to detect beads in mediastinal lymph nodes (data not shown) as it was previously reported Jakubzick et al., 2008. The relative contribution of monocytes and DCs in the transport of particles to the draining lymph node would require to analyse at later time points as performed by Jakubzick et al. The meaning of clearance has been adjusted according to reviewer suggestion, We discussed the point of particle transport and citations of the past studies have been added (subsection “Lung monocyte-derived cells survey both airways and vascular routes whereas DC survey only airways” and Discussion).

Minor points:

1) Both Ly6C^high^ and Ly6C^low^ monocytes express surface CX3CR1, as shown by Geissmann et al. The authors should state this more clearly in their Introduction.

This is now clarified in the manuscript (Introduction).

2) Parabiosis: To ensure that parabiosis led to efficient mixing in the parabionts, the authors should provide the number of mixed circulating T cells, B cells, neutrophils and monocytes in parabionts.

The following information is now provided in the Materials and methods section:

“Blood leukocyte chimerism was evaluated 2 weeks after surgery, and showed T cell chimerism at 50 ± 10%. In these settings, monocyte chimerism from B6 to MacBlue × Cx3cr1^gfp/+^ was around 37 ± 14% and MacBlue × Cx3cr1^gfp/+^ to B6 was reduced to 14 ± 13%.”

We suspected a strong competitivity of C57Bl6 non-fluorescent monocytes against MacBluexCx3cr1gfp/+ monocyte. Indeed, the chimerism of fluorescent monocytes in B6 host was variable and low despite a good chimerism of B6 monocytes in MacBluexCx3cr1^gfp/+^ host (Figure 11). For that reason we did not performed quantitative analysis of the monocyte infiltration but only studied the phenotype and the behaviour of lung monocytes.

Author response image 2.Ly6C^high^ monocyte chimerism in the blood of C57Bl6 and MacBluexCx3cr1^gfp/+^ (Tg) mice at 1 month after parabiosis. Results are expressed in % of Ly6C^high^ monocytes from the donor. Each dot represents one host mouse.**DOI:**
http://dx.doi.org/10.7554/eLife.07847.024

3) The movies could be rendered more aesthetically pleasing and easier to follow if more time was spent in processing them for display to readers.

Several specific indications have been drawn in each movie to make them easier to follow.

4) It might be worthwhile to indicate in Figure 2 that the difference in CD115 staining is likely due to cleavage in response to digestive enzymes used.

We specifically addressed this point in the new Figure 9 and in Figure 10.

Indeed Collagen IV digestion of the lung and the blood reduced the level of CD115 and some other markers. However we confirmed that lung monocytes downregulate CD115 expression independently of enzyme digestion. We discussed this point in the manuscript (please see the subsection “Lung monocyte-derived cells are located at the interface between blood and airways” and the Discussion).

5) The number of alveolar macrophages, interstitial macrophages and DC in the lung should also be quantified.

This quantification is now provided Figure 7 and discussed in the subsection “Interstitial monocyte-derived cells localized in the alveolar space whereas lung dendritic cells preferentially localized nearby airways”.